# Attention Implements the Fisher Geometry of Exponential Families

**Bodie Rubacher** [1]

## Abstract

Softmax attention normalizes scores, and Bayes' rule normalizes log prior plus log likelihood. For finite latent symbols with exponential-family observations, we show that one attention head can implement the Bayes posterior and posterior means exactly, and that the posteriors representable by a single head are precisely log-linear. The standard exponential-family duality identity rewrites the likelihood as a negative Bregman divergence in mean/sufficient-statistic space; our attention-specific contribution is to use this identity to characterize when Bayes-aligned attention admits one globally shared quadratic metric, proving that this happens exactly when the dual potential is quadratic. When curvature varies, we give a multi-head local-curvature atlas with approximation and head-count bounds, and we extend the picture to in-context estimation through plug-in consistency, finite-sample stability, and an optimizer-agnostic converse from excess log-loss to approximate key-subspace alignment. Controlled Gaussian, Bernoulli, and Poisson ICE diagnostics illustrate these regimes, while the exact theorems remain scoped to finite discrete latent classes and suggest testable, not universal, predictions for larger learned transformers.

## 1. Introduction

Transformers often exhibit in-context learning (ICL): given a prompt of examples followed by a query, the model solves the new query without parameter updates. This behavior appears in large language models (Brown et al., 2020) and has been analyzed in controlled settings such as in-context linear regression (Garg et al., 2022; Akyürek et al., 2023; Mahankali et al., 2024; Zhang et al., 2024). A Bayesian perspective has also emerged, interpreting ICL as implicit posterior inference over task parameters (Xie et al., 2022; Ahn et al., 2023; von Oswald et al., 2023; Panwar et al., 2024), and extending beyond function learning to representation learning and algorithm selection (Bai et al., 2023; Park et al., 2025).

A harder statistical variant is in-context estimation (ICE): estimate a latent symbol/class from a new noisy observation using additional i.i.d. labeled samples from the same hidden context. In wireless detection with unknown channel state, Kunde et al. (2025) proved that a single-layer softmax-attention transformer can asymptotically implement the MMSE estimator for a Gaussian channel, and that the optimal query-key metric equals the inverse noise covariance. This connects classical communication theory (Goldsmith, 2005) and learned receivers/channel estimation (Neumann et al., 2018; Caciularu and Burshtein, 2020; Ait Aoudia and Hoydis, 2022) to modern transformer inference.

These results raise two questions: (i) for which generative models can attention represent Bayes-optimal estimators, and (ii) what statistical geometry is encoded by learned attention similarity? We answer both at the level of representational capacity for discrete exponential-family observation models, and we formalize when a single global "metric" suffices versus when multiple heads are necessary.

The basic intuition is simple. Attention weights are normalized scores. Bayes' rule also normalizes scores: log prior plus log likelihood. In exponential families, the likelihood score is linear in the sufficient statistic, so it has the same bilinear form as query–key attention. Convex duality then says that this linear score is equivalently a negative Bregman divergence in mean coordinates. Thus the metric seen by a Bayes-aligned head is not generally Euclidean or Mahalanobis; it is the local Fisher/Bregman geometry of the observation model.

**Contributions.**

- **Bayes-Aligned Single-Head Representation.** For any fixed context in a finite discrete exponential-family model, a single attention head exactly reproduces the Bayes posterior and posterior mean by using sufficient statistics as queries, natural parameters as keys, and log-prior-minus-log-partition biases (Theorem 1). Con-

---

[1]Independent Researcher. Correspondence to: Bodie Rubacher <brubacher.1@gmail.com>.

*Proceedings of the 43rd International Conference on Machine Learning*, Seoul, South Korea. PMLR 306, 2026. Copyright 2026 by the author(s).

versely, the single-head posterior class is exactly the log-linear / discrete exponential-family class (Proposition 1).

- **Attention-Specific Geometry and Sharp Quadratic Boundary.** The Bregman/Fisher identities used in Section 4 are standard exponential-family convex duality. Our use of them is attention-specific: they identify the score computed by Bayes-aligned attention and yield a sharp boundary. A single globally shared quadratic query–key metric is Bayes-correct on an open mean-parameter region if and only if $A^*$ is quadratic there (Corollary 1).

- **Multi-Head Curvature Atlas with Rates.** When curvature varies, we give an explicit local-quadratic atlas approximation for $D_{A^*}$ (Proposition 3), convert logit error to posterior error via softmax stability (Lemma 2), and relate the number of heads to covering numbers / metric entropy of the task-relevant mean-parameter region (Proposition 4 and Appendix A).

- **ICE Beyond Gaussians.** We organize ICE into a ladder of regimes: fixed-context representability, oracle dictionary ICE, non-oracle plug-in consistency, and finite-sample posterior control (Theorems 3–5). This generalizes the Gaussian-channel story to arbitrary regular exponential-family observation models under the stated assumptions.

- **Necessity and Learning-to-Geometry Bridge.** Appendix C gives an exact parameter-level converse under a richness condition on $T(Y)$. In the main text, Corollary 2 composes the appendix results with the population cross-entropy decomposition to show that near-optimal log-loss forces approximate alignment of the learned key subspace with the Bayes log-odds subspace.

## 2. Setup

### 2.1. Discrete Exponential-Family Observation Models

Let $\mathcal{X}$ be a finite set of latent symbols/classes and let $\mathcal{Y}$ be the observation space. For a context $\theta$, we assume an exponential-family likelihood of the form

$$p_\theta(y|x) = \exp(\eta_\theta(x)^\top T(y) - A_\theta(x) - C_\theta(y)), \quad (1)$$

where $T(y) \in \mathbb{R}^k$ is a sufficient statistic and $\eta_\theta(x) \in \mathbb{R}^k$ is a context-dependent natural parameter. We assume a prior $P_X$ on $\mathcal{X}$ and treat $\theta$ as fixed for now.

Then Bayes' rule gives

$$P(x|y, \theta) \propto \exp\Big( \underbrace{\log P_X(x) - A_\theta(x)}_{b_\theta(x)} + \eta_\theta(x)^\top T(y)\Big).$$

$$(2)$$

Background: (Amari and Nagaoka, 2000; Wainwright and Jordan, 2008).

### 2.2. Single-Head Attention

Given a query $q \in \mathbb{R}^k$, keys $\{k(x)\}_{x \in \mathcal{X}} \subset \mathbb{R}^k$ and values $\{v(x)\}_{x \in \mathcal{X}} \subset \mathbb{R}^m$, a single head produces weights and output

$$\alpha(x; q) = \frac{e^{q^\top k(x)}}{\sum_{x'} e^{q^\top k(x')}} \quad (3)$$

$$\text{Attn}(q) = \sum_x \alpha(x; q)v(x). \quad (4)$$

Biases $b(x)$ can be implemented by augmenting $q, k$ with a constant feature dimension.

## 3. Attention as a Bayes-Optimal Estimator

### 3.1. Exact Bayes Representation for a Fixed Context

We build a dictionary attention layer with one key-value per $x \in \mathcal{X}$:

$$q(y) = T(y),$$
$$k(x) = \eta_\theta(x), \quad v(x) = g(x),$$
$$b_\theta(x) = \log P_X(x) - A_\theta(x).$$

Define logits $s_\theta(x; y) = q(y)^\top k(x) + b_\theta(x)$ and weights $\alpha_\theta(x; y) = \text{softmax}_x\, s_\theta(x; y)$.

**Theorem 1** (Attention implements Bayes for a fixed context). *For the model (1), with the above choice of $q, k, v$ and bias $b_\theta$, we have for all $y$:*

$$\alpha_\theta(x; y) = P(x|y, \theta)$$
$$\sum_{x \in \mathcal{X}} \alpha_\theta(x; y)g(x) = \mathbb{E}[g(x)|y, \theta].$$

*Proof.* Comparing (2) with the logits

$$s_\theta(x; y) = \eta_\theta(x)^\top T(y) + \log P_X(x) - A_\theta(x)$$

shows that

$$\log P(x|y, \theta) = s_\theta(x; y) - \log Z_\theta(y),$$

where the normalization term $-\log Z_\theta(y)$ is independent of $x$. Since the Softmax function is invariant to additive shifts (terms constant in $x$), we have

$$\alpha_\theta(x; y) = P(x|y, \theta).$$

This completes the proof. □

## 3.2. Which Posteriors Can a Single Head Represent?

**Proposition 1** (Log-linear posteriors $\Leftrightarrow$ exponential-family likelihoods). *Let $\mathcal{X}$ be finite with prior $P_X$. A posterior $P(x|y)$ is representable as*

$$P(x|y) = \frac{\exp(\phi(x)^\top \psi(y) + b(x))}{\sum_{x'} \exp(\phi(x')^\top \psi(y) + b(x'))} \tag{5}$$

*for some $\phi : \mathcal{X} \to \mathbb{R}^k$, $\psi : \mathcal{Y} \to \mathbb{R}^k$ and $b : \mathcal{X} \to \mathbb{R}$ if and only if there exists a likelihood of exponential-family form*

$$p(y|x) = \exp(\phi(x)^\top \psi(y) - A(x) - C(y))$$

*inducing $P(x|y)$ via Bayes' rule.*

*Proof.* Bayes' rule gives

$$P(x|y) \propto P_X(x) p(y|x).$$

If (5) holds then $p(y|x)$ must be proportional to

$$\frac{\exp(\phi(x)^\top \psi(y) + b(x))}{P_X(x)},$$

and the remaining $y$-only normalization yields $C(y)$. The converse is immediate from (2). $\qquad\square$

**Exact and Approximate Converses (Necessity).** Proposition 1 characterizes the set of posteriors representable by a single head. Appendix C strengthens this to a parameter-level converse under a mild richness condition on $T(Y)$: if a single head matches the Bayes posterior on the support, then its keys, queries, and biases must align with the exponential-family natural parameters and sufficient statistic up to softmax gauge and a nullspace term, yielding an intrinsic dimension lower bound. The same appendix also provides approximate converses connecting KL error to squared log-odds error and to approximate key-subspace alignment.

**Gaussian Special Case (No "Drop the Quadratic" Needed).** For $y = Hx + z$ with $z \sim \mathcal{N}(0, \Sigma_z)$,

$$\log p(y|x, H) = y^\top \Sigma_z^{-1} Hx - \frac{1}{2} x^\top H^\top \Sigma_z^{-1} Hx + y\text{-only}.$$

Thus Bayes logits are

$$s(x; y) = y^\top \Sigma_z^{-1} Hx + \log P_X(x) - \frac{1}{2} x^\top H^\top \Sigma_z^{-1} Hx.$$

Attention recovers Bayes with $q(y) = \Sigma_z^{-1} y$, $k(x) = Hx$ and an $x$-dependent bias. Under additional assumptions the quadratic term may be constant in $x$ (as in (Kunde et al., 2025)), but this is not required for Bayes representation.

# 4. Geometry: From Bregman Divergence to Fisher Precision

Section 4 explains what similarity Bayes-aligned attention is computing. While the Gaussian story is "Mahalanobis distance," general exponential families are governed by convex duality and Bregman geometry. The convex-conjugacy, Bregman, and Fisher identities in Sections 4.1–4.2 are standard background in exponential-family information geometry (Amari and Nagaoka, 2000; Banerjee et al., 2005; Wainwright and Jordan, 2008); the new results below use these identities to characterize attention scores, the global quadratic boundary, and multi-head atlas approximations.

## 4.1. Convex Duality and Bregman Form of the Likelihood

Write the standard exponential-family form (suppressing $\theta$ for clarity)

$$p(y|\eta) = h(y) \exp(\eta^\top T(y) - A(\eta)), \tag{6}$$

and define the mean map $\mu(\eta) = \nabla A(\eta) = \mathbb{E}_\eta[T(Y)]$ (regular/minimal case). Let $\mathcal{M} := \nabla A(\mathcal{U}) \subset \mathbb{R}^k$ denote the mean-parameter space, where $\mathcal{U}$ is the natural-parameter domain.

Let $A^*$ be the convex conjugate $A^*(t) = \sup_\eta \{\eta^\top t - A(\eta)\}$ and define the Bregman divergence

$$D_{A^*}(t, \mu) = A^*(t) - A^*(\mu) - \nabla A^*(\mu)^\top (t - \mu). \tag{7}$$

**Theorem 2** (Log-likelihood as negative Bregman divergence). *For $\mu = \nabla A(\eta)$,*

$$\eta^\top T(y) - A(\eta) = A^*(T(y)) - D_{A^*}(T(y), \mu). \tag{8}$$

*Consequently, for the latent-symbol model,*

$$P(x|y, \theta) \propto P_X(x) \exp(-D_{A^*}(T(y), \mu_\theta(x))), \tag{9}$$

*up to $y$-only terms that cancel in normalization.*

*Proof.* Legendre duality gives $A^*(\mu) = \eta^\top \mu - A(\eta)$ and $\nabla A^*(\mu) = \eta$. Plug into the definition of $D_{A^*}$ and rearrange. $\qquad\square$

**Interpretation.** Equation (9) says Bayes classification is a soft nearest-neighbor rule in the intrinsic Bregman geometry of the exponential family, with prior offset $\log P_X(x)$.

## 4.2. Fisher as Local Quadratic Approximation; Gaussian Is Globally Quadratic

The Fisher information is $F(\eta) = \nabla^2 A(\eta) = \mathrm{Cov}_\eta[T(Y)]$. In mean coordinates $\mu = \nabla A(\eta)$, the dual Hessian is

$$G(\mu) = \nabla^2 A^*(\mu) = F(\eta)^{-1}.$$

A Taylor/integral remainder identity yields the local approximation

$$D_{A^*}(t, \mu) = \frac{1}{2}(t - \mu)^\top G(\mu)(t - \mu) + o(||t - \mu||^2). \quad (10)$$

In Gaussian location families with covariance $\Sigma_z$, $D_{A^*}$ is exactly quadratic and $G(\mu) \equiv \Sigma_z^{-1}$ globally, recovering the Mahalanobis/precision metric of (Kunde et al., 2025). For Bernoulli, curvature varies as $1/(p(1 - p))$ so no single global metric matches all regimes.

### 4.3. When Does a Single Shared Metric Suffice? A Sharp Boundary

A single-head "metric" view posits a fixed matrix $W$ such that Bayes scores depend on $(t - \mu)$ through $\frac{1}{2}(t - \mu)^\top W(t - \mu)$ up to $t$-only and $x$-only terms. The following proposition shows this is only possible when the dual potential $A^*$ is quadratic.

**Proposition 2** (Affine $\nabla A^* \Leftrightarrow$ quadratic $A^*$). *Let $K \subset \mathcal{M}$ be convex open and assume $A^* \in C^2(K)$. If there exist $W \in \mathbb{R}^{k \times k}$ and $a \in \mathbb{R}^k$ such that*

$$\nabla A^*(\mu) = W\mu + a \quad \forall \mu \in K, \quad (11)$$

*then on $K$,*

$$A^*(\mu) = \frac{1}{2}\mu^\top W\mu + a^\top \mu + c \quad \text{for some constant } c, \quad (12)$$

*and the Bregman divergence is globally Mahalanobis:*

$$D_{A^*}(t, \mu) = \frac{1}{2}(t - \mu)^\top W(t - \mu) \quad \forall t, \mu \in K. \quad (13)$$

*Conversely, any $A^*$ of the form (12) satisfies (11) and (13).*

*Proof.* Fix $\mu_0 \in K$ and write $\gamma(s) = \mu_0 + s(\mu - \mu_0)$. By the fundamental theorem of calculus,

$$A^*(\mu) - A^*(\mu_0) = \int_0^1 \nabla A^*(\gamma(s))^\top (\mu - \mu_0) ds.$$

Using $\nabla A^*(\gamma(s)) = W\gamma(s) + a = W\mu_0 + a + sW(\mu - \mu_0)$ yields

$$A^*(\mu) - A^*(\mu_0) = (W\mu_0 + a)^\top (\mu - \mu_0)$$
$$+ \frac{1}{2}(\mu - \mu_0)^\top W(\mu - \mu_0)$$

which rearranges to (12) for an appropriate constant $c$. For quadratic $A^*(u) = \frac{1}{2}u^\top Wu + a^\top u + c$ one has $\nabla A^*(u) = Wu + a$ and direct substitution into the Bregman definition gives $D_{A^*}(t, \mu) = \frac{1}{2}(t - \mu)^\top W(t - \mu)$. The converse direction is immediate. $\square$

**Corollary 1** (Sharp boundary for globally shared quadratic attention metrics). *Let $K \subset \mathcal{M}$ be convex and open, and assume $A^* \in C^2(K)$. For $t, \mu \in K$, define the Bayes likelihood score in mean coordinates by*

$$\ell_{A^*}(t, \mu) := -D_{A^*}(t, \mu).$$

*Then the following are equivalent.*

1. *There exist a fixed matrix $W \in \mathbb{R}^{k \times k}$ and functions $b, c : K \to \mathbb{R}$ such that, for all $t, \mu \in K$,*

$$\ell_{A^*}(t, \mu) = t^\top W\mu + b(\mu) + c(t).$$

   *Equivalently, up to terms depending only on the query statistic $t$ and terms absorbable into class/key biases, Bayes scoring is representable by one shared quadratic attention metric $-\frac{1}{2}(t - \mu)^\top W(t - \mu)$.*

2. *The dual potential is quadratic on $K$: there exist $a \in \mathbb{R}^k$ and $c_0 \in \mathbb{R}$ such that*

$$A^*(u) = \frac{1}{2}u^\top Wu + a^\top u + c_0, \quad u \in K.$$

   *Equivalently, $\nabla^2 A^*(u) \equiv W$ on $K$.*

*When these conditions hold,*

$$D_{A^*}(t, \mu) = \frac{1}{2}(t - \mu)^\top W(t - \mu), \quad t, \mu \in K.$$

*Consequently, if $G(\mu) := \nabla^2 A^*(\mu)$ varies over $K$, then no single fixed Mahalanobis query–key metric can be globally Bayes-correct on $K$.*

*Proof.* The Bregman identity gives

$$-D_{A^*}(t, \mu) = \nabla A^*(\mu)^\top t + A^*(\mu) - \nabla A^*(\mu)^\top \mu - A^*(t).$$

If this score also equals $t^\top W\mu + b(\mu) + c(t)$ on the open set $K$, subtracting the two expressions and comparing two choices $\mu_1, \mu_2$ shows that

$$t^\top \{\nabla A^*(\mu_1) - \nabla A^*(\mu_2) - W(\mu_1 - \mu_2)\} + C(\mu_1, \mu_2) = 0$$

for all $t \in K$. Since $K$ is open, the coefficient of $t$ vanishes, so $\nabla A^*(\mu) - W\mu$ is constant on $K$. Proposition 2 then implies that $A^*$ is quadratic and that $D_{A^*}$ is globally Mahalanobis. Conversely, direct substitution of a quadratic $A^*$ gives

$$-D_{A^*}(t, \mu) = t^\top W\mu - \frac{1}{2}\mu^\top W\mu - \frac{1}{2}t^\top Wt,$$

where the last term is query-only and the middle term is absorbable into the class/key bias. $\square$

Corollary 1 is the precise sense in which the Gaussian/Mahalanobis case is exceptional. A single head can still represent Bayes logits for a general exponential family by using keys $\nabla A^*(\mu_x) = \eta_x$ and biases $\log P_X(x) - A(\eta_x)$. The obstruction is not to single-head Bayes representation, but to interpreting the score as one globally shared quadratic metric on mean/sufficient-statistic space.

### 4.4. Multi-Head Attention as a Local-Curvature Atlas

When $\nabla^2 A^*$ varies over the relevant region, we approximate the Bregman divergence by patching together local quadratic models.

**Lemma 1** (Integral remainder for Bregman divergences). *Assume $A^* \in C^2$ on a convex set containing the segment $\{\mu + s(t - \mu) : s \in [0,1]\}$. Then*

$$D_{A^*}(t, \mu) = \int_0^1 (1-s)(t-\mu)^\top$$
$$\nabla^2 A^*(\mu + s(t-\mu))(t-\mu)\, ds. \quad (14)$$

*Proof.* Let $\varphi(s) = A^*(\mu + s(t - \mu))$. Then $\varphi'(s) = \nabla A^*(\mu + s(t-\mu))^\top (t-\mu)$ and $\varphi''(s) = (t-\mu)^\top \nabla^2 A^*(\mu + s(t-\mu))(t-\mu)$. Taylor's theorem with integral remainder gives

$$\varphi(1) = \varphi(0) + \varphi'(0) + \int_0^1 (1-s)\varphi''(s)ds,$$

which is exactly (14). $\square$

**Proposition 3** (Curvature atlas approximation with explicit constants). *Let $K \subset \mathcal{M}$ be compact and assume $\nabla^2 A^*$ is $L$-Lipschitz (operator norm) on the convex hull of $K$:*

$$||\nabla^2 A^*(u) - \nabla^2 A^*(v)|| \le L||u-v|| \quad \forall u, v \in conv(K).$$

*Let $\{\mu_h\}_{h=1}^H \subset K$ form a $\rho$-net of $K$ and define $W_h = \nabla^2 A^*(\mu_h)$. For each $\mu \in K$ choose an index $h(\mu)$ with $||\mu - \mu_{h(\mu)}|| \le \rho$. Then for all $t, \mu \in K$,*

$$|D_{A^*}(t,\mu) - \frac{1}{2}(t-\mu)^\top W_{h(\mu)}(t-\mu)|$$
$$\le \frac{L}{2}\rho||t-\mu||^2 + \frac{L}{6}||t-\mu||^3. \quad (15)$$

*Proof.* Set $v = t - \mu$. From Lemma 1,

$$D_{A^*}(t,\mu) - \frac{1}{2}v^\top W_{h(\mu)}v$$
$$= \int_0^1 (1-s)v^\top \big(\nabla^2 A^*(\mu + sv)$$
$$- W_{h(\mu)}\big)v\, ds.$$

Using $|v^\top M v| \le ||M||||v||^2$ and the triangle inequality,

$$||\nabla^2 A^*(\mu + sv) - W_{h(\mu)}|| \le Ls||v|| + L\rho.$$

Therefore

$$|\text{LHS}| \le \int_0^1 (1-s)||v||^2 (Ls||v|| + L\rho)\, ds,$$

which integrates to the claimed bound. $\square$

**Interpreting the Two Terms in** (15). The term $\frac{L}{2}\rho||t-\mu||^2$ is the atlas discretization error: it comes from using the chart Hessian $W_{h(\mu)} = \nabla^2 A^*(\mu_{h(\mu)})$ at a net point $\mu_{h(\mu)}$ within distance $\rho$ of $\mu$. The term $\frac{L}{6}||t-\mu||^3$ is an intrinsic Taylor remainder due to curvature variation along the segment $\{\mu + s(t-\mu) : s \in [0,1]\}$; it becomes negligible in the small-noise regime $||t-\mu|| \ll 1$.

**Implementability by Attention.** For fixed $W_h$ the approximate Bayes logit $-\frac{1}{2}(t-\mu_x)^\top W_h(t-\mu_x)$ expands as

$$t^\top W_h\mu_x - \frac{1}{2}\mu_x^\top W_h\mu_x + t\text{-only},$$

so an attention head can implement it with query $q = t$, key $k = W_h\mu_x$, and an $x$-dependent bias $-\frac{1}{2}\mu_x^\top W_h\mu_x$; the $t$-only term cancels in the softmax over $x$.

**Remark 1** (Atlas as Piecewise-Linear Approximation; Implementability Variants). Proposition 3 can be read as approximating the nonlinear map $\mu \mapsto \nabla A^*(\mu)$ by a piecewise-linear family $\mu \mapsto W_h\mu$ over a $\rho$-net. The resulting logits have the attention-compatible bilinear+bias form (up to $t$-only terms), i.e. a dot product between a query feature $t$ and a head-specific key $W_h\mu_x$ plus an $x$-dependent bias. Exact realization of the class-dependent chart selection $\mu_x \mapsto h(\mu_x)$ is most naturally expressed as a logit-level mixture-of-heads (hard or soft gating).

**Proposition 4** (Head count scaling via covering numbers). *Assume $K \subset \mathbb{R}^k$ lies in a Euclidean ball of radius $R_K$. For any $\rho > 0$ there exists a $\rho$-net of $K$ with*

$$H \le (1 + \frac{2R_K}{\rho})^k. \quad (16)$$

*If additionally $||t-\mu|| \le R$ for all candidate means $\mu \in K$ (a noise-radius regime), then Proposition 3 gives a uniform logit error bound $\frac{L}{2}\rho R^2 + \frac{L}{6}R^3$. The posterior TV error is at most $\tanh(\epsilon(\rho)/2)$ (Lemma 2). Ignoring the intrinsic cubic floor (small when $R$ is small), achieving $\epsilon(\rho) \le \epsilon$ is possible with*

$$H \le (\frac{LR^2 R_K}{\epsilon})^k. \quad (17)$$

*Proof.* A standard volumetric argument yields (16) (cover a ball by radius balls). The logit and TV statements follow by substituting $||t-\mu|| \le R$ into (15) and using Lemma 2. $\square$

**Remark 2 (Metric-Entropy/Effective-Dimension Form).**
More generally, letting $N(K, \rho)$ denote the minimal $\rho$-covering number of the relevant mean-parameter region $K$, the same construction yields an atlas with $H = N(K, \rho)$ heads and uniform logit error $\epsilon(\rho) \leq \frac{L}{2}\rho R^2 + \frac{L}{6}R^3$ in the noise-radius regime $||t - \mu|| \leq R$, hence posterior TV error $\leq \tanh(\epsilon(\rho)/2)$ by Lemma 2. If $N(K, \rho) \leq C_K \rho^{-d_{eff}}$ for some intrinsic dimension $d_{eff}$, then (ignoring the cubic floor) achieving logit error $\leq \epsilon$ requires $H = O(C_K(LR^2/\epsilon)^{d_{eff}})$. A full statement and proof are in Appendix A.

**Optimization Caveat.** Propositions 3–4 are approximation-theoretic existence results. They show that if the task-relevant mean-parameter region $K$ can be covered by local charts $\{\mu_h\}_{h=1}^{H}$, then choosing $W_h = \nabla^2 A^*(\mu_h)$ and assigning each candidate mean $\mu$ to a nearby chart $h(\mu)$ yields a controlled local-quadratic approximation to $D_{A^*}(t, \mu)$. Thus the head-count bounds should be read as saying how many local quadratic charts are sufficient to approximate the Bayes geometry to a given accuracy, once an appropriate cover and assignment are available.

These results do not by themselves imply that standard gradient-based training of a multi-head attention layer will recover the optimal chart centers, Hessians, or assignment map. Such a learning guarantee would require additional assumptions on the training distribution, the parameterization of the heads, the gating or routing mechanism, and the optimization landscape. Accordingly, the curvature-atlas interpretation is a representational and geometric principle, not an end-to-end optimization theorem.

# 5. In-Context Estimation (ICE) Beyond Gaussians

Let $\theta \sim P_\theta$ be a latent context. Conditional on $\theta$, draw i.i.d. pairs $(x_t, y_t)_{t \geq 1}$ with $x_t \sim P_X$ and $y_t \sim p_\theta(\cdot|x_t)$. Given a prompt $\{(x_t, y_t)\}_{t=1}^{n}$ and a query observation $y_{n+1}$, we wish to estimate $x_{n+1}$.

The results below form a regime ladder. Theorem 1 is a fixed-context representability theorem: it assumes the context-specific natural parameters are known. Theorem 3 moves to an oracle dictionary ICE setting where the prompt contains samples from the fixed context but the logits still use oracle parameters. Theorems 4–5 then replace oracle parameters by plug-in estimates and give consistency and finite-sample posterior control. This ladder is not a complete theory of large-model ICL; it isolates the statistical primitive and the additional estimation error introduced by learning the context from examples.

## 5.1. Oracle Dictionary ICE (Expressivity)

Consider the oracle estimator

$$\hat{g}_n^{\text{oracle}}(y_{n+1}) = \frac{N_n(y_{n+1})}{D_n(y_{n+1})}, \tag{18}$$

$$N_n(y) = \sum_{t=1}^{n} g(x_t) \exp(\eta_\theta(x_t)^\top T(y) - A_\theta(x_t)),$$

$$D_n(y) = \sum_{t=1}^{n} \exp(\eta_\theta(x_t)^\top T(y) - A_\theta(x_t)).$$

**Theorem 3** (Oracle ICE asymptotics (dictionary regime)).
*For any bounded* $g$,

$$\hat{g}_n^{oracle}(y_{n+1}) \to \mathbb{E}[g(x_{n+1})|y_{n+1}, \theta] \tag{19}$$

*almost surely as* $n \to \infty$.

*Proof.* Group terms by symbol: $N_x^{(n)} = \sum_{t=1}^{n} \mathbb{I}\{x_t = x\}$. Then (18) equals a ratio of sums over $x$ with coefficients $N_x^{(n)}$. By the strong law, $N_x^{(n)}/n \to P_X(x)$ almost surely, yielding the Bayes posterior-mean formula. $\square$

## 5.2. Non-Oracle Plug-In ICE

The oracle theorem assumes access to $\eta_\theta(x)$ and $A_\theta(x)$. In regular minimal exponential families, the mean map $\mu(\eta) = \nabla A(\eta)$ is invertible, so we can estimate $\eta_\theta(x)$ from labeled samples via empirical sufficient statistics.

Let $N_x^{(n)} = \sum_{t=1}^{n} \mathbb{I}\{x_t = x\}$ and define the class-conditional empirical mean statistic

$$\overline{T}_x^{(n)} = \frac{1}{N_x^{(n)}} \sum_{t \leq n: x_t = x} T(y_t), \quad N_x^{(n)} \geq 1. \tag{20}$$

Assume $P_X(x) > 0$ for all $x \in \mathcal{X}$, so $N_x^{(n)} \to \infty$ almost surely.

Define the plug-in natural parameter estimator

$$\hat{\eta}_x^{(n)} = (\nabla A)^{-1}(\overline{T}_x^{(n)}), \tag{21}$$

and plug-in logits for a query $y_{n+1}$ with $t_{n+1} = T(y_{n+1})$:

$$\hat{s}_n(x; y_{n+1}) = \log P_X(x) + (\hat{\eta}_x^{(n)})^\top t_{n+1} - A(\hat{\eta}_x^{(n)}). \tag{22}$$

$$\hat{P}_n(x|y_{n+1}) = \text{softmax}_x \, \hat{s}_n(x; y_{n+1}). \tag{23}$$

(The $y$-only term cancels under normalization.)

## 5.3. Softmax Stability (Finite-$n$ Control)

**Lemma 2** (Softmax stability under bounded relative logit error). *Let* $p = softmax(s)$ *and* $\tilde{p} = softmax(s + \delta)$ *be distributions on a finite set* $\mathcal{X}$, *where* $p(x) \propto e^{s(x)}$ *and*

$\tilde{p}(x) \propto e^{s(x)+\delta(x)}$. *Define the logit-range perturbation* $\Delta := \max_{x \in \mathcal{X}} \delta(x) - \min_{x \in \mathcal{X}} \delta(x)$. *Then*

$$||p - \tilde{p}||_{TV} \leq \tanh(\Delta/4). \tag{24}$$

*In particular, if* $|\delta(x)| \leq \epsilon$ *for all* $x$, *then* $\Delta \leq 2\epsilon$ *and hence* $||p - \tilde{p}||_{TV} \leq \tanh(\epsilon/2)$. *Consequently, for any bounded* $g : \mathcal{X} \to \mathbb{R}$,

$$|\mathbb{E}_{\tilde{p}}[g] - \mathbb{E}_p[g]| \leq 2||g||_\infty \tanh(\Delta/4)$$
$$\leq 2||g||_\infty \tanh(\epsilon/2). \tag{25}$$

*Proof.* Softmax is shift-invariant, so $\text{softmax}(s + \delta) = \text{softmax}(s + \delta - c\mathbf{1})$ for any constant $c$. Choose $c = \frac{1}{2}(\max_x \delta(x) + \min_x \delta(x))$ so that the shifted perturbation (still denoted $\delta$) satisfies $\delta(x) \in [-\Delta/2, \Delta/2]$. Let $w(x) := e^{\delta(x)} \in [a, b]$ with $a = e^{-\Delta/2}$ and $b = e^{\Delta/2}$, and let $m := \mathbb{E}_p[w] = \sum_x p(x)w(x)$. Then $\tilde{p}(x) = \frac{p(x)w(x)}{m}$ and

$$||p - \tilde{p}||_{TV} = \frac{1}{2} \sum_x p(x) \left| \frac{w(x)}{m} - 1 \right| = \frac{1}{2} \mathbb{E}_p \left[ \left| \frac{w}{m} - 1 \right| \right].$$

For fixed $m \in [a, b]$, the function $w \mapsto \left| \frac{w}{m} - 1 \right|$ is convex on $[a, b]$, so by an extremal/convexity argument the expectation is maximized when $w$ puts all mass on the endpoints $\{a, b\}$. Thus it suffices to consider $w \in \{a, b\}$ with $P(w = a) = \theta$ and $P(w = b) = 1 - \theta$. A direct calculation gives

$$||p - \tilde{p}||_{TV} = \frac{\theta(1-\theta)(b-a)}{\theta a + (1-\theta)b},$$

which is maximized at $\theta^* = \frac{b - \sqrt{ab}}{b - a}$, yielding

$$||p - \tilde{p}||_{TV} = \frac{\sqrt{b} - \sqrt{a}}{\sqrt{b} + \sqrt{a}}.$$

With $a = e^{-\Delta/2}$ and $b = e^{\Delta/2}$ this equals $\tanh(\Delta/4)$. The bound on $|\mathbb{E}_{\tilde{p}}[g] - \mathbb{E}_p[g]|$ follows from $|\mathbb{E}_{\tilde{p}}[g] - \mathbb{E}_p[g]| \leq \sum_x |g(x)||p(x) - \tilde{p}(x)| \leq 2||g||_\infty ||p - \tilde{p}||_{TV}$. $\square$

**A Complementary Direction via KL (Appendices).** Lemma 2 bounds posterior error given a uniform logit perturbation. Appendix C provides a converse-style control in the other direction under an interiority condition: small KL divergence implies small centered log-odds error, and (in expectation) small log-odds error forces approximate alignment of the learned key subspace with the Bayes log-odds subspace. Appendix D connects population cross-entropy optimality to expected KL, so near-optimal log-loss implies posterior closeness.

**Theorem 4** (Plug-in ICE consistency). *Assume a regular minimal exponential family with continuous $(\nabla A)^{-1}$ on the relevant mean-parameter region and* $P_X(x) > 0$ *for all*

$x \in \mathcal{X}$. *Then conditional on any fixed $\theta$, for a fresh query* $y_{n+1} \sim p_\theta(\cdot | x_{n+1})$,

$$\hat{P}_n(\cdot | y_{n+1}) \xrightarrow[n \to \infty]{a.s.} P(\cdot | y_{n+1}, \theta). \tag{26}$$

*Consequently, for any bounded g,*

$$\sum_x g(x)\hat{P}_n(x | y_{n+1}) \to \mathbb{E}[g(x_{n+1}) | y_{n+1}, \theta] \tag{27}$$

*almost surely.*

*Proof.* By the strong law, $\overline{T}_x^{(n)} \to \mu_\theta(x) := \mathbb{E}[T(Y) | x, \theta]$ a.s. By continuity of $(\nabla A)^{-1}$, $\hat{\eta}_x^{(n)} \to \eta_\theta(x)$ a.s. Continuity of $\eta \mapsto \eta^\top t_{n+1} - A(\eta)$ implies $\hat{s}_n(x; y_{n+1}) \to s_\theta(x; y_{n+1})$ a.s., and softmax is continuous on finite alphabets. The posterior-mean convergence follows by bounded convergence. $\square$

### 5.4. Finite-$n$ Plug-In ICE Rates

We now give an explicit nonasymptotic bound on the plug-in posterior error. Fix $\theta$ and abbreviate $\eta_x := \eta_\theta(x)$, $\mu_x := \nabla A(\eta_x) = \mathbb{E}[T(Y) | X = x, \theta]$. For a fresh query $y$ write $t := T(y)$. The true Bayes logits are $s_\theta(x; y) = \log P_X(x) + \eta_x^\top t - A(\eta_x)$, and the plug-in logits are $\hat{s}_n(x; y) = \log P_X(x) + (\hat{\eta}_x^{(n)})^\top t - A(\hat{\eta}_x^{(n)})$ with $\hat{\eta}_x^{(n)} = (\nabla A)^{-1}(\overline{T}_x^{(n)})$.

**Assumption 1** (Uniform Fisher bounds on a relevant region). There exist constants $0 < m \leq M < \infty$ and a convex set $\mathcal{U} \subset \mathbb{R}^k$ such that for all $\eta \in \mathcal{U}$,

$$mI \preceq \nabla^2 A(\eta) \preceq MI, \tag{28}$$

and for all $x \in \mathcal{X}$ both $\eta_x \in \mathcal{U}$ and $\hat{\eta}_x^{(n)} \in \mathcal{U}$.

**Assumption 2** (Noise-radius for the query statistic). There exists $R > 0$ such that for the query $y$ under consideration, $\max_{x \in \mathcal{X}} ||t - \mu_x|| \leq R$.

**Lemma 3** (Inverse mean map is Lipschitz under a Fisher lower bound). *Under Assumption 1, for any $\mu, \mu' \in \nabla A(\mathcal{U})$,*

$$||\nabla A^*(\mu) - \nabla A^*(\mu')|| \leq \frac{1}{m} ||\mu - \mu'||. \tag{29}$$

*In particular, on the event $||\overline{T}_x^{(n)} - \mu_x|| \leq r$ we have $||\hat{\eta}_x^{(n)} - \eta_x|| \leq \frac{r}{m}$.*

*Proof.* On $\nabla A(\mathcal{U})$, $\nabla^2 A^*(\mu) = (\nabla^2 A(\eta))^{-1}$ with $\mu = \nabla A(\eta)$. Assumption 1 gives $||\nabla^2 A^*(\mu)|| \leq 1/m$. Apply the mean value theorem to $\nabla A^*$. $\square$

Define the parameter error $\Delta_x := \hat{\eta}_x^{(n)} - \eta_x$.

**Lemma 4** (Plug-in logit error). *Under Assumptions 1-2, if* $||\Delta_x|| \le \alpha$ *for all* $x$, *then*

$$\max_{x \in \mathcal{X}} |\hat{s}_n(x; y) - s_\theta(x; y)| \le R\alpha + \frac{M}{2}\alpha^2. \quad (30)$$

*Proof.* Write

$$\hat{s}_n(x; y) - s_\theta(x; y) = \Delta_x^\top t - (A(\eta_x + \Delta_x) - A(\eta_x)).$$

Using the Bregman identity $A(\eta_x + \Delta_x) - A(\eta_x) = \mu_x^\top \Delta_x + D_A(\eta_x + \Delta_x, \eta_x)$ where $D_A(\eta', \eta) := A(\eta') - A(\eta) - \nabla A(\eta)^\top(\eta' - \eta) \ge 0$, we obtain

$$\hat{s}_n - s_\theta = \Delta_x^\top(t - \mu_x) - D_A(\eta_x + \Delta_x, \eta_x).$$

Thus $|\hat{s}_n - s_\theta| \le ||\Delta_x|| ||t - \mu_x|| + D_A(\eta_x + \Delta_x, \eta_x)$. Under Assumption 1 (M-smoothness), $D_A(\eta_x + \Delta_x, \eta_x) \le \frac{M}{2}||\Delta_x||^2$. Under Assumption 2, $||t - \mu_x|| \le R$. Combine and use $||\Delta_x|| \le \alpha$. $\square$

Let $r_n := \max_{x \in \mathcal{X}} ||\overline{T}_x^{(n)} - \mu_x||$ and $N_{min} := \min_{x \in \mathcal{X}} N_x^{(n)}$.

**Theorem 5** (Finite-n plug-in ICE posterior bound). *Under Assumptions 1-2, on the event* $r_n \le r$ *we have the uniform logit error bound*

$$\max_x |\hat{s}_n(x; y) - s_\theta(x; y)| \le \epsilon(r) := \frac{R}{m}r + \frac{M}{2m^2}r^2. \quad (31)$$

*Therefore, by Lemma 2,*

$$||\hat{P}_n(\cdot|y) - P(\cdot|y, \theta)||_{TV} \le \tanh(\epsilon(r)/2). \quad (32)$$

*Proof.* If $r_n \le r$ then Lemma 3 gives $||\Delta_x|| \le r/m$ for all $x$. Substitute $\alpha = r/m$ into Lemma 4 to obtain the logit bound $\epsilon(r)$ then apply Lemma 2. $\square$

**Corollary 2** (Excess cross-entropy implies approximate key-subspace alignment). *Fix a context* $\theta$ *and let* $\mathcal{X} = \{x_0, x_1, \ldots, x_{C-1}\}$ *be finite. Let* $\pi_\theta(x|y) = P(x|y, \theta)$ *be the Bayes posterior and let a single attention head produce* $\alpha_\phi(x|y) = \text{softmax}_x(k_\phi(x)^\top q_\phi(y) + b_\phi(x))$. *Define the population cross-entropy*

$$\mathcal{L}_\theta(\phi) := \mathbb{E}_{(X,Y) \sim P_\theta}[-\log \alpha_\phi(X|Y)]$$

*and the Bayes risk* $H_\theta(X|Y) := \mathbb{E}[-\log \pi_\theta(X|Y)]$. *Fix reference class* $x_0$. *Let* $E_\theta \in \mathbb{R}^{(C-1) \times k}$ *have rows* $(\eta_\theta(x_i) - \eta_\theta(x_0))^\top$, *let* $K_\phi \in \mathbb{R}^{(C-1) \times d}$ *have rows* $(k_\phi(x_i) - k_\phi(x_0))^\top$, *and let* $P_{K_\phi}$ *be the orthogonal projector onto* $\text{col}(K_\phi)$. *Let* $\Sigma_T := \text{Cov}_{P_\theta}(T(Y))$.

*Assume* $\lambda_{min}(\Sigma_T) > 0$ *and the posterior interiority condition: for* $P_\theta$-*almost every* $y$,

$$\min_x \pi_\theta(x|y) \ge \rho, \qquad \min_x \alpha_\phi(x|y) \ge \rho$$

*for some* $\rho > 0$. *If* $\mathcal{L}_\theta(\phi) - H_\theta(X|Y) \le \varepsilon$, *then*

$$\mathbb{E}_Y\left[\text{KL}(\pi_\theta(\cdot|Y) \| \alpha_\phi(\cdot|Y))\right] \le \varepsilon,$$

*and the reference log-odds error*

$$e_\phi(y)_i := \log \frac{\alpha_\phi(x_i|y)/\alpha_\phi(x_0|y)}{\pi_\theta(x_i|y)/\pi_\theta(x_0|y)}, \quad i = 1, \ldots, C - 1,$$

*satisfies*

$$\mathbb{E}_Y \|e_\phi(Y)\|_2^2 \le \frac{2C}{\rho}\varepsilon.$$

*Consequently,*

$$\left\|(I - P_{K_\phi})E_\theta\right\|_F^2 \le \frac{2C}{\rho \, \lambda_{min}(\Sigma_T)}\varepsilon.$$

*Proof.* The population cross-entropy decomposition gives

$$\mathcal{L}_\theta(\phi) - H_\theta(X|Y) = \mathbb{E}_Y[\text{KL}(\pi_\theta(\cdot|Y) \| \alpha_\phi(\cdot|Y))].$$

Thus the excess-risk assumption gives the expected KL bound. Lemma 6, applied pointwise with $n = C$, converts KL to squared reference log-odds error under the interiority condition. Theorem 7 then converts mean-squared log-odds error to key-subspace alignment, giving the stated bound. This is optimizer-agnostic: it does not prove that gradient descent reaches a near-optimal solution, but any single-head solution that is near-optimal in population log-loss is geometrically constrained in this sense. $\square$

**Prompt-Level Concentration.** The event $r_n \le r$ in Theorem 5 can be controlled by standard concentration inequalities for bounded or sub-Gaussian sufficient statistics, together with Chernoff bounds for the random class counts. We give the explicit high-probability forms in Appendix E.

## 6. Experiments (Synthetic ICE)

We validate four qualitative predictions in controlled synthetic ICE settings: (i) performance improves toward a Bayes-oracle baseline as prompt length grows, (ii) in Gaussian ICE the learned metric aligns with the Bayes precision up to scale, (iii) non-Gaussian curvature variation creates a mismatch for one shared quadratic surrogate, and (iv) in a curvature-heterogeneous Poisson regime, a fixed-scale multi-head atlas can outperform a single shared quadratic surrogate in the low-shot regime (Appendix B).

Following (Kunde et al., 2025), we train minimal attention models on episodic data.

**Gaussian Learned Geometry.** A $C$-class model $y|x = c \sim \mathcal{N}(\mu_c, \Sigma_z)$ with episode-specific means $\{\mu_c\}$ and fixed $\Sigma_z$. The Bayes score has the shared quadratic form predicted by Corollary 1. We learn an SPD metric $W = LL^\top$ and score

with $s_c(y) = y^\top W \hat{\mu}_c - \frac{1}{2} \hat{\mu}_c^\top W \hat{\mu}_c$. To test whether training recovers the Bayes geometry, let $P_\star := \Sigma_z^{-1}$ and choose the best scale $c^\star = \langle W, P_\star \rangle_F / \|P_\star\|_F^2$ matching $P_\star$ to $W$. Across five seeds, the learned metric tracks $P_\star$ up to scale: the sorted-eigenvalue correlation is $0.960 \pm 0.005$, and the scaled relative Frobenius error $\|W - c^\star P_\star\|_F / \|c^\star P_\star\|_F$ is $0.355 \pm 0.005$. Thus trained minimal attention recovers the predicted precision-like metric in this controlled Gaussian diagnostic.

**Bernoulli.** Independent features $y_j | x = k \sim$ Bernoulli$(p_{k,j})$ with episode-specific $p_k$. Accuracy improves substantially with prompt length, but a gap to Bayes remains, consistent with curvature variation and Proposition 2.

**Poisson Topic Scale (Non-Gaussian Curvature Heterogeneity).** Appendix B gives a controlled Poisson ICE task where the dual Hessian $G(\lambda) = \nabla^2 A^*(\lambda) = \text{diag}(1/\lambda)$ varies strongly with an episode-level scale. In this setting, a fixed-scale $H = 8$ atlas yields substantial low-shot gains over a single shared quadratic surrogate (e.g., $n = 1 : 0.428 \to 0.684$), matching the curvature-atlas prediction when curvature varies.[1]

The full Poisson accuracy table is reported in Appendix B.

## 7. Related Work

Transformers as in-context learners for simple function classes, especially linear regression, are studied in (Garg et al., 2022; Akyürek et al., 2023; Mahankali et al., 2024; Zhang et al., 2024). Bayesian interpretations of ICL as implicit inference over task parameters appear in (Xie et al., 2022; Ahn et al., 2023; von Oswald et al., 2023; Panwar et al., 2024). Recent work explores transformers as approximate Bayesian inference engines over richer model classes, including full posteriors and empirical Bayes formulations (Reuter et al., 2025; Teh et al., 2025). In-context estimation for wireless detection is introduced in (Kunde et al., 2025), building on classical communication models (Goldsmith, 2005) and learned receiver/channel-estimation methods (Neumann et al., 2018; Caciularu and Burshtein, 2020; Ait Aoudia and Hoydis, 2022).

Our bilinear exponential-family structure is related to exponential-family embeddings (Rudolph et al., 2016) and recent exponential-family attention models (Wibisono and Wang, 2025). Other geometric and optimization views of attention include one-sided entropic optimal-transport interpretations of scaled dot-product attention (Litman, 2025) and mirror-descent analyses of softmax attention dynamics and max-margin token selection (Julistiono et al., 2026). Re-

cent work also studies the information geometry induced by softmax output distributions for probing and steering learned representations (Park et al., 2026). Our focus is different and complementary: in a generative exponential-family setting, the likelihood itself induces the Bregman/Fisher geometry that a Bayes-aligned attention score must compute.

## 8. Discussion and Limitations

**Scope.** The exact representability, converse, and stability results are proved for finite discrete $\mathcal{X}$. Finiteness enters through the class softmax, finite log-odds vectors, dictionary attention over candidate symbols, and total-variation control of finite posterior distributions. The theory characterizes Bayes-aligned scoring in this clean model class; it does not assert that arbitrary large transformers universally learn Fisher geometry.

**Continuous Latent Spaces.** Appendix F gives the density-level analogue. In brief, the finite softmax over $x$ becomes an integral-normalized posterior density with logits $s_\theta(x; y) = \eta_\theta(x)^\top T(y) + \log p_X(x) - A(\eta_\theta(x))$, and the Bregman form becomes $\alpha_\theta(dx|y) \propto p_X(x) \exp\{-D_{A^*}(T(y), \mu_\theta(x))\} \nu(dx)$. A theorem-level extension would need common domination, measurability, finite nonzero normalizers, regularity of $A, A^*$, tail/integrability control, continuum stability bounds, and function-space log-density-ratio converses.

**Scaling and Depth.** The testable prediction is conditional: when a learned component is near Bayes-optimal for an approximately exponential-family conditional task, near-constant curvature should favor one shared quadratic metric, while varying curvature should favor multiple local charts or nonquadratic scoring. Depth may reparameterize sufficient-statistic space, refine charts, or compose atlas approximations, but we do not prove an optimization or deep-stack theorem here.

## 9. Conclusion

For finite discrete exponential-family observation models, one attention head can represent Bayes posteriors and posterior means exactly. The induced Bayes geometry is Bregman/Fisher: a shared quadratic metric is globally exact only when $A^*$ is quadratic. When curvature varies, the theory motivates multiple local charts and gives ICE representability, plug-in, stability, and near-optimal-log-loss-to-geometry results. The scoped takeaway is practical: use one metric in near-constant-curvature regimes; otherwise expect local heads or nonquadratic scoring to be necessary.

---

[1] In Appendix B (Poisson), we restrict to diagonal $W$ for scalability.

## Impact Statement

This work clarifies when it is principled to treat softmax attention as an inference primitive, and when doing so is structurally misleading. On the positive side, we give conditions under which Bayes-aligned attention induces an intrinsic Bregman/Fisher geometry on mean/sufficient-statistic space, and we provide concrete design guidance: (i) an attention-centric boundary showing that an *exact* single shared quadratic query–key metric can only be globally correct when the dual potential is quadratic (hence the geometry is globally Mahalanobis), and (ii) a quantitative multi-head "curvature atlas" approximation with explicit error terms and head-count scaling via covering numbers / metric entropy.

These results can improve the reliability of attention-based estimators and learned receivers in settings where in-context estimation is natural (e.g., communications and other noisy sensing tasks), because they separate regimes where a single learned metric is justified from regimes where curvature variation forces multiple heads (or non-quadratic scoring) to avoid systematic mismatch. We also provide ICE-specific statistical packaging: oracle asymptotics, plug-in consistency, and finite-$n$ posterior control via stability bounds, making explicit what can (and cannot) be guaranteed at realistic prompt lengths.

Potential broader impacts are dual-use. Stronger Bayes-like attention mechanisms can benefit safety-critical perception and robust estimation, but can also strengthen surveillance, interception, or monitoring capabilities in communications contexts. A central misuse risk is *over-trust and over-interpretation*: treating attention similarity as a globally meaningful metric outside the regimes where our boundary and atlas conditions apply. We therefore view the boundary (Proposition 2) and the explicit approximation/stability guarantees (Propositions 3–4; Theorems 4–5) as guardrails: they identify when "metric" interpretations are warranted, and when a deployment should instead expect curvature-driven failure modes and require additional auditing or constraint.

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

## A. Metric-Entropy View of Head Count (Full Statement)

**Metric-Entropy View of Head Count.** Proposition 4 uses an ambient-dimension volumetric bound for the size of a $\rho$-net. More generally, the required number of heads is controlled by the metric entropy (covering numbers) of the relevant mean-parameter region $K$.

**Corollary 3** (Head count via metric entropy / effective dimension). *Assume the conditions of Proposition 3. Fix $R > 0$ such that the noise-radius regime holds: $||t - \mu|| \le R$ for all $t$ under consideration and all $\mu \in K$. Let*

$$N(K, \rho) := \min\Big\{ H : \exists \{\mu_h\}_{h=1}^{H} \subset K \text{ with } K \subset \bigcup_{h=1}^{H} B(\mu_h, \rho) \Big\}$$

*denote the minimal Euclidean $\rho$-covering number of $K$. Then for any $\rho > 0$, there exists an $H$-head curvature-atlas attention mechanism with $H = N(K, \rho)$, whose Bayes-logit approximation error is uniformly bounded by*

$$\epsilon(\rho) := \frac{L}{2} \rho R^2 + \frac{L}{6} R^3. \tag{33}$$

*Consequently, by softmax stability (Lemma 2), the induced posterior satisfies*

$$||\hat{P}(\cdot|t) - P(\cdot|t)||_{TV} \le \tanh(\epsilon(\rho)/2). \tag{34}$$

*In particular, for any target logit accuracy $\epsilon > \frac{L}{6} R^3$, choosing*

$$\rho_\epsilon := \frac{2}{LR^2} \left( \epsilon - \frac{L}{6} R^3 \right) \tag{35}$$

*yields $\epsilon(\rho_\epsilon) \le \epsilon$ and thus $H \le N(K, \rho_\epsilon)$. Moreover, if $K$ has metric-entropy growth $N(K, \rho) \le C_K \rho^{-d}$ for some intrinsic dimension $d$, then*

$$H = O\left( C_K \left( \frac{LR^2}{\epsilon} \right)^d \right). \tag{36}$$

*Proof.* Let $\{\mu_h\}_{h=1}^{H}$ be a $\rho$-net of $K$ of minimal size $H = N(K, \rho)$, and define $W_h = \nabla^2 A^*(\mu_h)$ as in Proposition 3. For each $\mu \in K$ choose $h(\mu)$ with $||\mu - \mu_{h(\mu)}|| \le \rho$. Proposition 3 bounds the Bregman-to-local-quadratic error by

$$\left| D_{A^*}(t, \mu) - \frac{1}{2}(t - \mu)^\top W_{h(\mu)}(t - \mu) \right| \le \frac{L}{2} \rho ||t - \mu||^2 + \frac{L}{6} ||t - \mu||^3.$$

Under $||t - \mu|| \le R$ this is at most $\epsilon(\rho)$. Interpreting this as a uniform logit perturbation and applying Lemma 2 gives the stated TV bound. Solving $\epsilon(\rho) \le \epsilon$ for $\rho$ yields $\rho_\epsilon$ and the head-count bound $H \le N(K, \rho_\epsilon)$. The entropy-growth specialization follows by plugging in $N(K, \rho) \le C_K \rho^{-d}$. $\square$

**Remark 3 (Recovering Proposition 4 and Geometric Interpretation).** Proposition 4 is recovered by the crude ambient-dimension bound $N(K, \rho) \le (1 + 2R_K/\rho)^k$ when $K \subset B(0, R_K) \subset \mathbb{R}^k$. This corollary makes explicit that the head count is governed by the metric entropy of $K$ at resolution $\rho \asymp \epsilon/(LR^2)$: i.e., heads act as charts in a local-curvature atlas over the statistical region explored by the task.

## B. Poisson Topic Scale ICE: A Non-Gaussian Curvature-Atlas Test

This note connects a controlled Poisson in-context estimation (ICE) experiment to the geometric framework of Sections 4-5 (Theorem 2, Proposition 2, Propositions 3-4), and reports the resulting accuracy curves. The goal is to test the qualitative claim: when $\nabla^2 A^*$ varies substantially over the relevant mean-parameter region, multiple heads (an atlas) can outperform a single shared quadratic surrogate, especially in the low-shot regime.

### B.1. Generative Model (Topic Scale Poisson Episodes)

Fix dimension $d$, number of classes $C$, and a global topic pool size $M$. We first sample a pool of $M$ topic vectors $\beta_m \in \Delta^{d-1}$, $\beta_m \sim \text{Dirichlet}(\alpha 1_d)$, $m = 1, \ldots, M$, and hold $\{\beta_m\}_{m=1}^{M}$ fixed across episodes.

An episode is generated as follows. For each class $c \in \{1, \ldots, C\}$:

1. Sample a topic index $z_c \sim \text{Unif}(\{1, \ldots, M\})$ and a scale $\log s_c \sim \text{Unif}(\log s_{min}, \log s_{max})$, $s_c \in [s_{min}, s_{max}]$.

2. Define the class mean (Poisson rate vector) $\lambda_c := s_c \beta_{z_c} \in \mathbb{R}_{>0}^d$.

Since $1^\top \beta_{z_c} = 1$, we have $1^\top \lambda_c = s_c$, i.e. the total rate equals the scale.

**Prompt/prototypes.** Given a prompt length $n$ per class, we draw $n$ i.i.d. labeled observations per class and compress them to a sufficient statistic. Coordinatewise, the sum of $n$ Poisson samples is Poisson, so we equivalently draw $C_c \sim \text{Poisson}(n\lambda_c)$ (coordinatewise), $\hat{\lambda}_c := \frac{C_c}{n}$. Thus $\hat{\lambda}_c$ is the plug-in MLE/empirical mean for $\lambda_c$.

**Query.** Sample a query label $x_q \sim \text{Unif}(\{1, \ldots, C\})$ and a query observation $Y \sim \text{Poisson}(\lambda_{x_q})$ (coordinatewise).

Given $Y$ and $\{\hat{\lambda}_c\}_{c=1}^{C}$, the task is to predict $x_q$.

### B.2. Oracle and Plug-In Bayes Logits (Poisson)

The independent-coordinate Poisson likelihood is $p(y|\lambda_c) = \prod_{j=1}^{d} \frac{\lambda_{c,j}^{y_j} e^{-\lambda_{c,j}}}{y_j!}$. Ignoring the $y$-only term $-\sum_j \log(y_j!)$, the (uniform-prior) Bayes-optimal logits are

$$s_c^{\text{oracle}}(y) = y^\top \log \lambda_c - 1^\top \lambda_c, \tag{37}$$

and the plug-in Bayes logits replace $\lambda_c$ by $\hat{\lambda}_c$:

$$s_c^{\text{plugin}}(y) = y^\top \log \hat{\lambda}_c - 1^\top \hat{\lambda}_c. \tag{38}$$

By the general plug-in ICE consistency results (Theorem 4), $s_c^{\text{plugin}}(Y)$ approaches the oracle logits as $n \to \infty$ under regularity. Since empirical Poisson prototypes may have zero coordinates at small $n$, any expression containing $\log \hat{\lambda}_c$ is evaluated with a positive regularized prototype, for example $\tilde{\lambda}_{c,j} = \max\{\hat{\lambda}_{c,j}, \tau\}$ for a fixed numerical floor $\tau > 0$, and with $\hat{\lambda}_c$ replaced by $\tilde{\lambda}_c$ in the logarithm. This keeps the plug-in logits inside the regular positive-mean region used by the theory.

### B.3. Bregman Geometry for Poisson (Instantiating Theorem 2)

Poisson is an exponential family with sufficient statistic $T(y) = y$ and natural parameter $\eta = \log \lambda$:

$$p(y|\eta) = h(y) \exp(\eta^\top y - A(\eta)), \quad A(\eta) = 1^\top e^\eta, \quad h(y) = \prod_{j=1}^{d} \frac{1}{y_j!}.$$

The mean map is $\mu(\eta) = \nabla A(\eta) = e^\eta = \lambda$, and the convex conjugate is

$$A^*(t) = \sup_\eta \{\eta^\top t - A(\eta)\} = \sum_{j=1}^{d} (t_j \log t_j - t_j) \quad (\text{for } t \in \mathbb{R}_{>0}^d).$$

**Convention for zeros.** We use the continuous extension $0 \log 0 := 0$, so $A^*(y)$ and $D_{A^*}(y, \lambda)$ are well-defined for $y \in \mathbb{Z}_{\geq 0}^d$ (terms with $y_j = 0$ contribute 0 in $y_j \log(y_j/\lambda_j)$).

Its gradient is $\nabla A^*(\mu) = \log \mu$. The associated Bregman divergence is

$$D_{A^*}(t, \mu) = A^*(t) - A^*(\mu) - \nabla A^*(\mu)^\top (t - \mu) = \sum_{j=1}^{d} \left[ t_j \log \frac{t_j}{\mu_j} - t_j + \mu_j \right], \tag{39}$$

the generalized KL divergence between nonnegative measures. Applying Theorem 2 (log-likelihood as negative Bregman divergence) with $t = T(y) = y$ and $\mu = \lambda_c$ yields

$$\eta^\top y - A(\eta) = A^*(y) - D_{A^*}(y, \lambda_c),$$

so Bayes classification is a soft nearest-neighbor rule in the intrinsic Poisson Bregman geometry:

$$P(x = c|y) \propto \exp(-D_{A^*}(y, \lambda_c)),$$

up to $y$-only normalization.

### B.4. Curvature Varies Strongly with Scale

In mean coordinates, the dual Hessian (Fisher precision in mean space) is

$$G(\mu) := \nabla^2 A^*(\mu) = \text{diag}\left(\frac{1}{\mu_1}, \ldots, \frac{1}{\mu_d}\right).$$

In this experiment $\mu = \lambda = s\beta$, so

$$G(\lambda) = \text{diag}\left(\frac{1}{s\beta_1}, \ldots, \frac{1}{s\beta_d}\right) = \frac{1}{s}\text{diag}\left(\frac{1}{\beta_1}, \ldots, \frac{1}{\beta_d}\right).$$

Hence the overall curvature magnitude scales like $1/s$. With $s \in [8, 512]$, curvature magnitudes vary by a factor of $64\times$ across episodes/classes, creating a regime where a single shared quadratic surrogate is a priori mismatched, motivating an atlas approximation as in Propositions 3-4.

### B.5. Attention-Form Quadratic Surrogates

Exact Bayes logits in exponential families take the attention form

$$s_c(y) = q(y)^\top k_c + b_c, \quad \text{with } q(y) = T(y), \ k_c = \eta_c, \ b_c = \log P_X(c) - A(\eta_c),$$

as in Theorem 1 / Eq. (2) of the main text. For Poisson, $q(y) = y$, $k_c = \log \lambda_c$, and $b_c = -\mathbf{1}^\top \lambda_c$ (plus prior).

Our learned models restrict to a quadratic family that is implementable as dot-product attention with an $x$-dependent bias:

$$s_c^{(W)}(y; \mu_c) := y^\top W \mu_c - \frac{1}{2}\mu_c^\top W \mu_c, \quad W \succeq 0 \text{ diagonal}. \tag{40}$$

When the same $W$ is used for all classes, this is equivalent (up to a $y$-only constant) to negative squared Mahalanobis distance:

$$y^\top W \mu_c - \frac{1}{2}\mu_c^\top W \mu_c = -\frac{1}{2}(y - \mu_c)^\top W (y - \mu_c) + \frac{1}{2}y^\top W y,$$

and the final term cancels in the softmax over $c$.

**Why We Do Not Use the Full Squared-Distance Logit When Heads Vary by Class.** If we allow per-class heads $W_{h(c)}$, the full distance score $-\frac{1}{2}(y - \mu_c)^\top W_{h(c)}(y - \mu_c)$ contains the term $-\frac{1}{2}y^\top W_{h(c)}y$, which is class-dependent when $h(c)$ varies. This introduces a query-only quadratic term that does not cancel across classes and is not of the standard attention logit form (dot product + bias). For this reason, we evaluate the atlas using the attention-consistent bilinear+bias logits.

### B.6. Single-Head vs. Fixed-Scale Atlas Models

We compare two trained models, both using the prototype $\mu_c = \hat{\lambda}_c$.

**Single-Head (Global Quadratic Surrogate).** Learn $w \in \mathbb{R}_{>0}^d$ and set $W = \text{diag}(w)$. Logits:

$$s_c^{\text{single}}(y) = y^\top W \hat{\lambda}_c - \frac{1}{2}\hat{\lambda}_c^\top W \hat{\lambda}_c.$$

**Multi-Head Fixed-Scale Atlas.** Learn $\{w_h\}_{h=0}^{H-1}$ with $W_h = \text{diag}(w_h)$. Assign a deterministic head to each class via its estimated scale

$$\hat{s}_c := 1^\top \hat{\lambda}_c \approx s_c, \quad h(c) := \left\lfloor H \cdot \frac{\log \hat{s}_c - \log s_{min}}{\log s_{max} - \log s_{min}} \right\rfloor \in \{0, \ldots, H-1\},$$

(with clipping at the endpoints). Logits:

$$s_c^{\text{atlas}}(y) = y^\top W_{h(c)} \hat{\lambda}_c - \frac{1}{2} \hat{\lambda}_c^\top W_{h(c)} \hat{\lambda}_c.$$

This is a controlled curvature atlas where head selection tracks the dominant curvature driver (scale), without any learned gating. It is therefore a test of the approximation principle, not an end-to-end optimization result showing that training discovers the routing map.

### B.7. Training and Evaluation Protocol

Both models are trained by episodic cross-entropy on freshly sampled episodes. During training, the prompt length $n$ is sampled from $\{1, 2, 4, 8, 16, 32\}$ uniformly. Evaluation reports accuracy at $n \in \{1, 2, 4, 8, 16, 32, 64, 128\}$.

**Hyperparameters for the Reported Run.** We used $d = 4096$, $C = 8$, $M = 2048$, $\alpha = 0.05$, $(s_{min}, s_{max}) = (8, 512)$, $H = 8$ heads, AdamW with learning rate $3 \times 10^{-3}$, weight decay $10^{-2}$, gradient clipping 1.0, training steps 15000, batch size 64. Each accuracy is computed over $n_{\text{eval}} = 3200$ query examples per $n$.

### B.8. Results

Table 1 reports accuracy (mean $\pm$ standard error over evaluation queries for the reported trained run) for: (i) the learned single-head quadratic surrogate, (ii) the learned fixed-scale atlas, (iii) plug-in Bayes, and (iv) oracle Bayes. These error bars are evaluation standard errors, not standard errors over independent training seeds.

*Table 1.* Poisson Topic Scale ICE accuracy vs. prompt length $n$ (per class), for a single-head quadratic surrogate and a fixed-scale multi-head atlas, compared against plug-in and oracle Bayes baselines.

| $n$ | SINGLE | ATLAS ($H{=}8$) | PLUG-IN | ORACLE |
|---|---|---|---|---|
| 1 | $.428 \pm .009$ | $.684 \pm .008$ | .938 | 1.00 |
| 2 | $.667 \pm .008$ | $.855 \pm .006$ | .958 | 1.00 |
| 4 | $.856 \pm .006$ | $.931 \pm .004$ | .984 | 1.00 |
| 8 | $.939 \pm .004$ | $.978 \pm .003$ | .996 | 1.00 |
| 16 | $.979 \pm .003$ | $.995 \pm .001$ | .996 | 1.00 |
| 32 | $.988 \pm .002$ | $.994 \pm .001$ | .998 | 1.00 |
| 64 | $.995 \pm .001$ | $.996 \pm .001$ | .998 | 1.00 |
| 128 | $.994 \pm .001$ | $.997 \pm .001$ | .999 | 1.00 |

**Effect Size in the Low-Shot Regime.** The atlas yields large low-shot gains (single seed run):

- $n = 1$: $0.428 \to 0.684$ (error reduction $\approx 45\%$).

- $n = 2$: $0.667 \to 0.855$ (error reduction $\approx 56\%$).

- $n = 4$: $0.856 \to 0.931$ (error reduction $\approx 52\%$).

The gap shrinks as $n$ increases and both learned surrogates approach near-ceiling accuracy.

**Head Usage / Specialization.** Across $n$, the atlas uses all $H = 8$ heads essentially uniformly (effective heads by exp-entropy $\approx 8.0$). For reference, Table 2 reports the geometric center of each log-scale bin and the mean diagonal weight magnitude learned for that head.

### B.9. Implications for the Main Theory

This experiment provides a concrete instantiation of the geometric picture:

*Table 2.* Atlas bin centers (in scale space) and learned per-head mean diagonal weights for the reported run.

| HEAD $h$ | BIN CENTER $\tilde{s}_h$ | MEAN WEIGHT $\frac{1}{d}\sum_{j=1}^{d}(w_h)_j$ |
|---|---|---|
| 0 | 10.4 | 2.186 |
| 1 | 17.4 | 1.303 |
| 2 | 29.3 | 0.792 |
| 3 | 49.4 | 0.475 |
| 4 | 83.0 | 0.295 |
| 5 | 139.6 | 0.338 |
| 6 | 234.8 | 0.537 |
| 7 | 394.8 | 0.656 |

1. **(Theorem 2 instantiated).** Poisson Bayes classification is soft nearest neighbor in the Poisson Bregman geometry, and oracle/plug-in logits are given by (37)–(38).

2. **(Curvature heterogeneity).** The dual Hessian $G(\lambda) = \nabla^2 A^*(\lambda) = \mathrm{diag}(1/\lambda)$ varies strongly with scale $s = 1^\top\lambda$, creating a regime where a single shared quadratic surrogate is mismatched, motivating an atlas (Propositions 3-4).

3. **(Attention implementability matters).** Using bilinear+bias logits ensures the scoring rule matches the standard attention logit structure (dot product + bias) and avoids class-dependent query-only quadratic terms that arise from naive per-class quadratic distances.

4. **(Empirical confirmation).** The fixed-scale atlas significantly improves low-shot performance while preserving near-ceiling behavior at larger $n$, consistent with the intuition that curvature-aware patching is most valuable when the relevant region is large / heterogeneous (low-shot), and yields diminishing returns as the estimation problem becomes easy.

5. **(Boundary/non-Gaussian expressivity).** Despite large gains, both quadratic surrogates are substantially below plug-in Bayes at $n = 1$–$2$, reflecting that Poisson Bayes logits depend on $\log\lambda$ (nonlinear in $\lambda$), so a quadratic/linear-key surrogate cannot be globally exact.

## C. Single-Head Attention and Bayes Posteriors: Exact and Approximate Converses

### C.1. Setup and Notation

Let $\mathcal{X}$ be finite with $n := |\mathcal{X}|$, and fix a prior $P_X$ on $\mathcal{X}$. Fix a context $\theta$ and consider the exponential-family likelihood (1),

$$p_\theta(y|x) = \exp(\eta_\theta(x)^\top T(y) - A_\theta(x) - C_\theta(y)), \quad T(y) \in \mathbb{R}^k,\ \eta_\theta(x) \in \mathbb{R}^k.$$

Define the $x$-dependent offset

$$\beta_\theta(x) := \log P_X(x) - A_\theta(x). \tag{41}$$

(Note $\beta_\theta(x)$ coincides with $b_\theta(x)$ in (2).) Then the Bayes posterior can be written as a softmax over logits

$$\pi_\theta(x|y) := P(x|y,\theta) = \mathrm{softmax}_x(s_\theta(x;y)), \quad s_\theta(x;y) := \eta_\theta(x)^\top T(y) + \beta_\theta(x). \tag{42}$$

A single attention head with a query map $q : \mathcal{Y} \to \mathbb{R}^d$, keys $k : \mathcal{X} \to \mathbb{R}^d$, and biases $b : \mathcal{X} \to \mathbb{R}$ produces

$$\alpha(x;y) = \mathrm{softmax}_x(u(x;y)), \quad u(x;y) := k(x)^\top q(y) + b(x). \tag{43}$$

(As in the main text, biases can be implemented by augmenting $(q, k)$ with a constant feature dimension.)

**Centering and Log-Odds.** Fix a reference class $x_0 \in \mathcal{X}$ and define centered differences

$$\Delta k(x) := k(x) - k(x_0) \in \mathbb{R}^d, \qquad \Delta\eta_\theta(x) := \eta_\theta(x) - \eta_\theta(x_0) \in \mathbb{R}^k,$$
$$\Delta b(x) := b(x) - b(x_0) \in \mathbb{R}, \qquad \Delta\beta_\theta(x) := \beta_\theta(x) - \beta_\theta(x_0) \in \mathbb{R}, \quad (x \neq x_0).$$

Define the log-odds vectors (indexed by $x \neq x_0$)

$$r_\theta(y) \in \mathbb{R}^{n-1}, \qquad\qquad [r_\theta(y)]_x := \log \frac{\pi_\theta(x|y)}{\pi_\theta(x_0|y)} = \Delta\eta_\theta(x)^\top T(y) + \Delta\beta_\theta(x), \qquad (44)$$

$$\hat{r}(y) \in \mathbb{R}^{n-1}, \qquad\qquad [\hat{r}(y)]_x := \log \frac{\alpha(x;y)}{\alpha(x_0;y)} = \Delta k(x)^\top q(y) + \Delta b(x). \qquad (45)$$

Let $E \in \mathbb{R}^{(n-1)\times k}$ and $K \in \mathbb{R}^{(n-1)\times d}$ be the matrices with rows

$$E_{x,\cdot} = \Delta\eta_\theta(x)^\top, \quad K_{x,\cdot} = \Delta k(x)^\top, \quad (x \neq x_0), \qquad (46)$$

and let $\beta_\Delta, b_\Delta \in \mathbb{R}^{n-1}$ have entries $\Delta\beta_\theta(x), \Delta b(x)$. Then

$$r_\theta(y) = E\,T(y) + \beta_\Delta, \quad \hat{r}(y) = K\,q(y) + b_\Delta. \qquad (47)$$

## C.2. Exact Converse: An Iff Characterization

**Lemma 5** (Softmax identifiability). *For $u, v \in \mathbb{R}^n$,*

$$softmax(u) = softmax(v) \iff u = v + c\mathbf{1} \text{ for some scalar } c \in \mathbb{R}.$$

*Proof.* Let $p = \text{softmax}(u)$ and $q = \text{softmax}(v)$. If $p = q$, then for all $i, j$,

$$\log \frac{p_i}{p_j} = u_i - u_j = \log \frac{q_i}{q_j} = v_i - v_j,$$

so $(u - v)_i = (u - v)_j$ for all $i, j$, i.e. $u - v = c\mathbf{1}$. Conversely, adding $c\mathbf{1}$ to logits does not change the softmax. $\square$

**Richness Assumption on $T(\mathcal{Y})$.** Assume that the affine hull of $\{T(y) : y \in \text{supp}(Y)\}$ is all of $\mathbb{R}^k$. Equivalently, there exist $y_0, \ldots, y_k$ such that $\{T(y_i) - T(y_0)\}_{i=1}^k$ spans $\mathbb{R}^k$. (If the affine hull has dimension $k' < k$, the results below hold with $k$ replaced by $k'$.)

**Theorem 6** (Exact converse / iff characterization). *Under the richness assumption, the following are equivalent:*

*(A) $\alpha(\cdot; y) = \pi_\theta(\cdot|y)$ for all $y$ in the support.*

*(B) There exist $M \in \mathbb{R}^{d\times k}$, $v \in \mathbb{R}^d$, a scalar $c \in \mathbb{R}$, and a (possibly $y$-dependent) nullspace term $n(y) \in \ker(K)$ such that:*

| | | |
|---|---|---|
| *(Key–natural-parameter alignment)* | $KM = E$, *equivalently* $\Delta\eta_\theta(x) = M^\top\Delta k(x)\ \forall x \neq x_0$; | (48) |
| *(Query–statistic alignment mod $\ker(K)$)* | $q(y) = M\,T(y) + v + n(y), \quad n(y) \in \ker(K)$; | (49) |
| *(Bias alignment)* | $b(x) = \beta_\theta(x) - v^\top k(x) + c \quad \forall x \in \mathcal{X}.$ | (50) |

*Equivalently, there exists $a \in \mathbb{R}^k$ such that $\eta_\theta(x) = M^\top k(x) + a$ for all $x \in \mathcal{X}$.*

*Proof.* **(A)$\Rightarrow$(B).** If $\alpha(\cdot; y) = \pi_\theta(\cdot|y)$, then by Lemma 5 there exists a scalar $c(y)$ such that

$$k(x)^\top q(y) + b(x) = \eta_\theta(x)^\top T(y) + \beta_\theta(x) + c(y) \quad \forall x \in \mathcal{X}.$$

Subtract the same equality at $x_0$ to eliminate $c(y)$:

$$\Delta k(x)^\top q(y) + \Delta b(x) = \Delta\eta_\theta(x)^\top T(y) + \Delta\beta_\theta(x) \quad \forall x \neq x_0.$$

In matrix form,

$$K\,q(y) + b_\Delta = E\,T(y) + \beta_\Delta, \quad \forall y. \qquad (51)$$

Fix $y$ and $y_0$ and subtract:

$$K(q(y) - q(y_0)) = E(T(y) - T(y_0)).$$

By richness, $\{T(y) - T(y_0)\}$ spans $\mathbb{R}^k$, hence $\mathrm{col}(E) \subseteq \mathrm{col}(K)$. Therefore there exists $M \in \mathbb{R}^{d \times k}$ such that $KM = E$, proving key–natural-parameter alignment.

Substitute $E = KM$ into the matrix equation:

$$K(q(y) - M\, T(y)) = \beta_\Delta - b_\Delta,$$

whose right-hand side is independent of $y$. Choose any $v \in \mathbb{R}^d$ such that $Kv = \beta_\Delta - b_\Delta$ (this is possible since $\beta_\Delta - b_\Delta \in \mathrm{col}(K)$). Then

$$K(q(y) - M\, T(y) - v) = 0,$$

so $q(y) = M\, T(y) + v + n(y)$ with $n(y) \in \ker(K)$, proving query–statistic alignment.

Finally, rearranging $Kv = \beta_\Delta - b_\Delta$ yields $b_\Delta = \beta_\Delta - Kv$, i.e. for all $x \neq x_0$,

$$\Delta b(x) = \Delta\beta_\theta(x) - v^\top \Delta k(x).$$

This implies there is a scalar $c$ such that $b(x) = \beta_\theta(x) - v^\top k(x) + c$ for all $x$, proving bias alignment.

To obtain the uncentered alignment $\eta_\theta(x) = M^\top k(x) + a$, set $a := \eta_\theta(x_0) - M^\top k(x_0)$ and use $\Delta\eta_\theta(x) = M^\top \Delta k(x)$.

**(B)⇒(A).** Assume the three alignments hold. Then for any $x \neq x_0$,

$$\Delta k(x)^\top q(y) + \Delta b(x) = \Delta k(x)^\top (M\, T(y) + v + n(y)) + \Delta\beta_\theta(x) - v^\top \Delta k(x)$$
$$= \Delta\eta_\theta(x)^\top T(y) + \Delta\beta_\theta(x),$$

since $\Delta k(x)^\top n(y) = 0$ for $n(y) \in \ker(K)$ and $\Delta k(x)^\top M\, T(y) = \Delta\eta_\theta(x)^\top T(y)$ by key alignment. Thus $\hat{r}(y) = r_\theta(y)$ for all $y$, hence the attention and Bayes posteriors have identical log-odds and therefore coincide. $\qquad\square$

**Corollary 4** (Intrinsic dimension lower bound). *Let $m_* := rank(E) = \dim span\{\eta_\theta(x) - \eta_\theta(x_0) : x \in \mathcal{X}\}$. If $\alpha(\cdot; y) = \pi_\theta(\cdot|y)$ for all $y$ under richness, then $rank(K) \geq m_*$, hence $d \geq m_*$.*

*Proof.* If $\alpha = \pi_\theta$ for all $y$, Theorem 6 yields $E = KM$ so $\mathrm{col}(E) \subseteq \mathrm{col}(K)$, implying $\mathrm{rank}(K) \geq \mathrm{rank}(E) = m_*$. $\qquad\square$

### C.3. Approximate Converse I: KL Controls Squared Log-Odds Error

**Log-Sum-Exp and Notation.** For $u \in \mathbb{R}^n$ define $F(u) := \log \sum_{i=1}^n e^{u_i}$. Then $\nabla F(u) = \mathrm{softmax}(u)$ and

$$\nabla^2 F(u) = \mathrm{Diag}(s) - ss^\top, \quad s = \mathrm{softmax}(u).$$

**Proposition 5** (KL is a Bregman divergence of log-sum-exp). *Let $p, q \in \Delta^{n-1}$ and define $a := \log p$, $b := \log q$. Then*

$$KL(p\|q) = D_F(b, a) := F(b) - F(a) - \nabla F(a)^\top (b - a).$$

*Proof.* Since $\sum_i p_i = \sum_i q_i = 1$, we have $F(\log p) = F(\log q) = 0$ and $\nabla F(\log p) = p$. Therefore

$$D_F(\log q, \log p) = 0 - 0 - p^\top(\log q - \log p) = \sum_i p_i \log \frac{p_i}{q_i} = KL(p\|q).$$

$\qquad\square$

**Proposition 6** (Strong convexity on the simplex interior). *Assume the interiority condition $\min_i p_i \geq \rho$ and $\min_i q_i \geq \rho$ for some $\rho > 0$. Let $\delta := \log q - \log p \in \mathbb{R}^n$, let $P := I - \frac{1}{n}\mathbf{1}\mathbf{1}^\top$ be the orthogonal projector onto $\mathbf{1}^\perp$, and set $\delta_c := P\delta$. Then*

$$KL(p\|q) \geq \frac{\rho}{2}\|\delta_c\|_2^2, \quad \text{equivalently } \|\delta_c\|_2^2 \leq \frac{2}{\rho}KL(p\|q).$$

*Proof.* By Proposition 5,

$$\mathrm{KL}(p\|q) = D_F(a + \delta, a), \quad a = \log p.$$

Define $g(t) := F(a + t\delta)$ for $t \in [0, 1]$. Then

$$g'(t) = \nabla F(a + t\delta)^\top \delta, \quad g''(t) = \delta^\top \nabla^2 F(a + t\delta)\delta.$$

The integral remainder identity gives

$$D_F(a + \delta, a) = \int_0^1 (1 - t)\delta^\top H_t \delta \, dt, \quad H_t := \nabla^2 F(a + t\delta). \tag{52}$$

Let $s_t := \mathrm{softmax}(a + t\delta) = \nabla F(a + t\delta)$, so $H_t = \mathrm{Diag}(s_t) - s_t s_t^\top$. For any $v \in \mathbb{R}^n$,

$$v^\top H_t v = \sum_{i=1}^n s_{t,i} v_i^2 - \left(\sum_{i=1}^n s_{t,i} v_i\right)^2 = \sum_{i=1}^n s_{t,i}(v_i - \mu_t)^2, \quad \mu_t := \sum_i s_{t,i} v_i. \tag{53}$$

**Step 1: Along the Logit Segment, $s_{t,i} \geq \rho$.** Since $a = \log p$ and $a + \delta = \log q$,

$$a_i + t\delta_i = (1 - t)\log p_i + t\log q_i \quad \Rightarrow \quad e^{a_i + t\delta_i} = p_i^{1-t} q_i^t.$$

Hence

$$s_{t,i} = \frac{p_i^{1-t} q_i^t}{\sum_j p_j^{1-t} q_j^t}.$$

By weighted AM–GM, $p_j^{1-t} q_j^t \leq (1 - t)p_j + tq_j$, so

$$\sum_j p_j^{1-t} q_j^t \leq (1 - t)\sum_j p_j + t\sum_j q_j = 1.$$

Therefore $s_{t,i} \geq p_i^{1-t} q_i^t \geq \rho$ for all $i$ and $t \in [0, 1]$.

**Step 2: Strong Convexity on $\mathbf{1}^\perp$.** Let $v \in \mathbf{1}^\perp$ so $\sum_i v_i = 0$. Using the variance identity and $s_{t,i} \geq \rho$,

$$v^\top H_t v = \sum_i s_{t,i}(v_i - \mu_t)^2 \geq \rho \sum_i (v_i - \mu_t)^2.$$

But

$$\sum_i (v_i - \mu_t)^2 = \sum_i v_i^2 - 2\mu_t \sum_i v_i + n\mu_t^2 = \|v\|_2^2 + n\mu_t^2 \geq \|v\|_2^2.$$

Thus $v^\top H_t v \geq \rho\|v\|_2^2$ for all $v \in \mathbf{1}^\perp$.

**Step 3: Remove the Constant-Logit Direction.** Since $H_t \mathbf{1} = 0$, adding any multiple of $\mathbf{1}$ to $\delta$ does not change $\delta^\top H_t \delta$. Hence $\delta^\top H_t \delta = (\delta_c)^\top H_t \delta_c$ where $\delta_c = P\delta \in \mathbf{1}^\perp$. Therefore

$$\delta^\top H_t \delta \geq \rho\|\delta_c\|_2^2,$$

and plugging into the integral remainder identity yields

$$\mathrm{KL}(p\|q) \geq \int_0^1 (1 - t)\rho\|\delta_c\|_2^2 \, dt = \frac{\rho}{2}\|\delta_c\|_2^2.$$

$\square$

**Indexing Convention for the KL/Log-Odds Bounds.** For the next results, fix an arbitrary ordering of $\mathcal{X}$ as $\{x_0, x_1, \ldots, x_{n-1}\}$ and identify vectors in $\mathbb{R}^n$ with components indexed by these elements; in particular, index 0 corresponds to the reference class $x_0$.

**Proposition 7** (Reference log-odds are controlled by centered logits). *Fix a reference index* $0$ *and define the linear operator* $D : \mathbb{R}^n \to \mathbb{R}^{n-1}$ *by*

$$(Dv)_i := v_i - v_0, \quad i = 1, \ldots, n - 1.$$

*Then for all* $v \in \mathbf{1}^\perp$,

$$||Dv||_2^2 \le n||v||_2^2.$$

*Moreover the constant* $n$ *is tight.*

*Proof.* Let $L := D^\top D$. One computes

$$L_{00} = n - 1, \quad L_{ii} = 1 \ (i \ge 1), \quad L_{0i} = L_{i0} = -1 \ (i \ge 1), \quad L_{ij} = 0 \ (i \ne j, \, i, j \ge 1).$$

Hence $||Dv||^2 = v^\top L v$. One checks:

- $L\mathbf{1} = 0$, so $0$ is an eigenvalue with eigenvector $\mathbf{1}$.

- If $u_0 = 0$ and $\sum_{i \ge 1} u_i = 0$, then $Lu = u$, giving eigenvalue 1 with multiplicity $n - 2$.

- For $w = (n - 1, -1, \ldots, -1)$, we have $w \in \mathbf{1}^\perp$ and $Lw = nw$, so $n$ is an eigenvalue.

Thus the largest eigenvalue of $L$ restricted to $\mathbf{1}^\perp$ is $n$, yielding $||Dv||^2 \le n||v||^2$ for $v \in \mathbf{1}^\perp$. Tightness follows by taking $v = w$. □

**Lemma 6** (KL $\Rightarrow$ squared log-odds error (pointwise)). *Assume* $\min_i p_i \ge \rho$ *and* $\min_i q_i \ge \rho$. *Let* $e \in \mathbb{R}^{n-1}$ *be the reference log-odds error*

$$e_i := \log \frac{q_i/q_0}{p_i/p_0} = (\log q_i - \log p_i) - (\log q_0 - \log p_0), \quad i = 1, \ldots, n - 1.$$

*Then*

$$||e||_2^2 \le \frac{2n}{\rho} KL(p\|q).$$

*Proof.* Let $\delta := \log q - \log p \in \mathbb{R}^n$. Then $e = D\delta$ with $D$ from Proposition 7. Since $D\mathbf{1} = 0$, we have $D\delta = D(P\delta) = D\delta_c$, where $\delta_c = P\delta \in \mathbf{1}^\perp$. Thus by Proposition 7,

$$||e||_2^2 = ||D\delta_c||_2^2 \le n||\delta_c||_2^2.$$

By Proposition 6, $||\delta_c||_2^2 \le \frac{2}{\rho} KL(p\|q)$. Combining gives

$$||e||_2^2 \le \frac{2n}{\rho} KL(p\|q).$$

□

## C.4. Approximate Converse II: Subspace Necessity from Squared Log-Odds Error

Let $\mathcal{Y}$ be any evaluation distribution on observations. Recall

$$r_\theta(Y) = E\, T(Y) + \beta_\Delta, \quad \hat{r}(Y) = K\, q(Y) + b_\Delta, \quad e(Y) = \hat{r}(Y) - r_\theta(Y).$$

Let $P_K$ denote the orthogonal projection in $\mathbb{R}^{n-1}$ onto $\mathrm{col}(K)$. Let $\Sigma_T := \mathrm{Cov}(T(Y))$ and assume $\lambda_{\min}(\Sigma_T) > 0$.

**Theorem 7** (From mean-squared log-odds error to key-subspace alignment). *With notation as above,*

$$||(I - P_K)E||_F^2 \le \frac{1}{\lambda_{\min}(\Sigma_T)} \mathbb{E}||e(Y)||_2^2.$$

*Consequently, under the interiority condition and writing* $\epsilon := \mathbb{E}[KL(\pi_\theta(\cdot|Y)\|\alpha(\cdot; Y))]$,

$$||(I - P_K)E||_F^2 \le \frac{2n}{\rho\, \lambda_{\min}(\Sigma_T)} \epsilon.$$

*Proof.* Define centered versions

$$r_c(Y) := r_\theta(Y) - \mathbb{E}[r_\theta(Y)], \quad \hat{r}_c(Y) := \hat{r}(Y) - \mathbb{E}[\hat{r}(Y)].$$

Since $\hat{r}(Y) = Kq(Y) + b_\Delta$, we have

$$\hat{r}_c(Y) = K(q(Y) - \mathbb{E}[q(Y)]) \in \text{col}(K) \quad \text{for all } Y.$$

Therefore, pointwise in $y$,

$$||(I - P_K)r_c(y)|| = \min_{z \in \text{col}(K)} ||r_c(y) - z|| \leq ||r_c(y) - \hat{r}_c(y)||.$$

Squaring and taking expectations,

$$\mathbb{E}||(I - P_K)r_c(Y)||_2^2 \leq \mathbb{E}||r_c(Y) - \hat{r}_c(Y)||_2^2. \tag{54}$$

But

$$r_c(Y) - \hat{r}_c(Y) = (r_\theta(Y) - \hat{r}(Y)) - \mathbb{E}[r_\theta(Y) - \hat{r}(Y)] = -(e(Y) - \mathbb{E}[e(Y)]).$$

Hence

$$\mathbb{E}||r_c(Y) - \hat{r}_c(Y)||_2^2 = \mathbb{E}||e(Y) - \mathbb{E}[e(Y)]||_2^2 = \mathbb{E}||e(Y)||_2^2 - ||\mathbb{E}[e(Y)]||_2^2 \leq \mathbb{E}||e(Y)||_2^2. \tag{55}$$

Combining gives

$$\mathbb{E}||(I - P_K)r_c(Y)||_2^2 \leq \mathbb{E}||e(Y)||_2^2. \tag{56}$$

Next, writing $\mu_T := \mathbb{E}[T(Y)]$, we have

$$r_c(Y) = E(T(Y) - \mu_T).$$

Therefore

$$\mathbb{E}||(I - P_K)r_c(Y)||_2^2 = \mathbb{E}[(T - \mu_T)^\top E^\top (I - P_K)E(T - \mu_T)] = \text{tr}(E^\top (I - P_K)E \Sigma_T).$$

Let $A := E^\top (I - P_K)E \succeq 0$. Since $\Sigma_T \succeq \lambda_{\min}(\Sigma_T)I$,

$$\text{tr}(A\Sigma_T) \geq \lambda_{\min}(\Sigma_T)\text{tr}(A) = \lambda_{\min}(\Sigma_T)\text{tr}(E^\top (I - P_K)E) = \lambda_{\min}(\Sigma_T)||(I - P_K)E||_F^2.$$

Thus

$$\lambda_{\min}(\Sigma_T)||(I - P_K)E||_F^2 \leq \mathbb{E}||(I - P_K)r_c(Y)||_2^2 \leq \mathbb{E}||e(Y)||_2^2,$$

which proves the first inequality.

For the KL-based consequence, apply Lemma 6 pointwise in $y$ with $p = \pi_\theta(\cdot|y)$ and $q = \alpha(\cdot; y)$, then take expectation:

$$\mathbb{E}||e(Y)||_2^2 \leq \frac{2n}{\rho}\mathbb{E}[\text{KL}(\pi_\theta(\cdot|Y)||\alpha(\cdot; Y))] = \frac{2n}{\rho}\epsilon.$$

$\square$

# D. Learning via Population Cross-Entropy Optimality

**Population Cross-Entropy Objective.** Let $(U, X) \sim P$ where $U$ denotes the available information (e.g., prompt + query) and $X$ is the target label. Let $q_\phi(\cdot|U)$ be any conditional distribution output by a model with parameters $\phi$. Define the population log-loss

$$\mathcal{L}(\phi) := \mathbb{E}[-\log q_\phi(X|U)]. \tag{57}$$

**Proposition 8** (Bayes posterior minimizes population cross-entropy)**.** *Let $P(\cdot|U)$ denote the true conditional distribution of $X$ given $U$. Then*

$$\mathcal{L}(\phi) = H(X|U) + \mathbb{E}[KL(P(\cdot|U)||q_\phi(\cdot|U))], \tag{58}$$

*where $H(X|U) := \mathbb{E}[-\log P(X|U)]$ is the conditional entropy. In particular, minimizing over all conditional distributions yields the unique minimizer $q(\cdot|U) = P(\cdot|U)$ a.s. and minimum value $H(X|U)$. Moreover, for any $\phi$ the excess risk over Bayes equals expected KL:*

$$\mathcal{L}(\phi) - H(X|U) = \mathbb{E}[KL(P(\cdot|U)||q_\phi(\cdot|U))]. \tag{59}$$

*Proof.* Write

$$\mathcal{L}(\phi) = \mathbb{E}[-\log P(X|U)] + \mathbb{E}\left[\log \frac{P(X|U)}{q_\phi(X|U)}\right] = H(X|U) + \mathbb{E}[\text{KL}(P(\cdot|U)\|q_\phi(\cdot|U))].$$

The remaining statements follow since KL is nonnegative and equals 0 iff the two distributions match a.s. $\qquad\square$

**Corollary 5** (Excess cross-entropy controls posterior error)**.** *Assume $X$ takes values in a finite set. By Pinsker's inequality, for each $U$,*

$$||P(\cdot|U) - q_\phi(\cdot|U)||_{TV} \leq \sqrt{\frac{1}{2}KL(P(\cdot|U)\|q_\phi(\cdot|U))}.$$

*Therefore,*

$$\mathbb{E}[||P(\cdot|U) - q_\phi(\cdot|U)||_{TV}] \leq \sqrt{\frac{1}{2}(\mathcal{L}(\phi) - H(X|U))}. \tag{60}$$

*Consequently, for any bounded g,*

$$\mathbb{E}[|\mathbb{E}_{q_\phi}[g(X)|U] - \mathbb{E}[g(X)|U]|] \leq 2||g||_\infty \sqrt{\frac{1}{2}(\mathcal{L}(\phi) - H(X|U))}. \tag{61}$$

# E. Concentration Details for Plug-In ICE

This appendix records the standard concentration statements used to instantiate the event $r_n \leq r$ in Theorem 5, where

$$r_n := \max_{x \in \mathcal{X}} \|\overline{T}_x^{(n)} - \mu_x\|, \qquad N_{min} := \min_{x \in \mathcal{X}} N_x^{(n)}.$$

**Bounded Sufficient Statistics.** Assume coordinate-wise boundedness: for all $j \in [k]$, $T_j(Y) \in [-B, B]$ almost surely. Conditional on the label sequence, equivalently on the counts $\{N_x^{(n)}\}$, a coordinate-wise Hoeffding bound and a union bound imply that, with probability at least $1 - \delta$ over the prompt observations,

$$r_n \leq B\sqrt{\frac{2k\log(2k|\mathcal{X}|/\delta)}{N_{min}}}.$$

**Sub-Gaussian Sufficient Statistics.** Assume for each $x$ that $T(Y) - \mu_x$ is $\sigma$-sub-Gaussian in $\mathbb{R}^k$. Then standard vector concentration gives, conditional on the counts and with probability at least $1 - \delta$,

$$r_n \lesssim \sigma\sqrt{\frac{k + \log(|\mathcal{X}|/\delta)}{N_{min}}},$$

up to a universal constant depending on the sub-Gaussian convention.

**Random class counts.** If $x_t \overset{i.i.d.}{\sim} P_X$ and $p_{min} := \min_x P_X(x) > 0$, then for any $\alpha \in (0,1)$,

$$\mathbb{P}\{N_x^{(n)} \leq (1-\alpha)nP_X(x)\} \leq \exp\{-\alpha^2 nP_X(x)/2\},$$

and hence, by a union bound,

$$\mathbb{P}\{N_{min} \leq (1-\alpha)np_{min}\} \leq |\mathcal{X}|\exp\{-\alpha^2 np_{min}/2\}.$$

Intersecting this count event with either concentration event above gives a high-probability finite-$n$ posterior bound through Theorem 5.

# F. Continuous-Latent Extension Sketch

This appendix expands the continuous-$\mathcal{X}$ discussion from Section 8. Let $(\mathcal{X}, \mathcal{A}, \nu)$ be a latent space with dominating measure $\nu$, let $p_X$ be a prior density, and suppose

$$p_\theta(y|x) = h(y) \exp\{\eta_\theta(x)^\top T(y) - A(\eta_\theta(x))\}.$$

If $x \mapsto \eta_\theta(x)$ is measurable and

$$0 < Z_\theta(y) := \int_{\mathcal{X}} p_X(x) \exp\{\eta_\theta(x)^\top T(y) - A(\eta_\theta(x))\} \, d\nu(x) < \infty,$$

then Bayes' rule gives the posterior density

$$p_\theta(x|y) = \frac{p_X(x) \exp\{\eta_\theta(x)^\top T(y) - A(\eta_\theta(x))\}}{Z_\theta(y)}.$$

If the exponential family is regular and minimal and $\mu_\theta(x) = \nabla A(\eta_\theta(x))$, then Legendre duality gives

$$\eta_\theta(x)^\top T(y) - A(\eta_\theta(x)) = A^*(T(y)) - D_{A^*}(T(y), \mu_\theta(x)).$$

The $A^*(T(y))$ term is independent of $x$ and cancels in the posterior normalizer, yielding

$$p_\theta(x|y) = \frac{p_X(x) \exp\{-D_{A^*}(T(y), \mu_\theta(x))\}}{\int_{\mathcal{X}} p_X(x') \exp\{-D_{A^*}(T(y), \mu_\theta(x'))\} \, d\nu(x')}.$$

Thus the finite dictionary softmax is formally replaced by a continuum softmax / normalized kernel over latent keys indexed by $x \in \mathcal{X}$.

A full theorem would require at least the following assumptions. First, all relevant priors and posteriors should be dominated by a common measure so that densities and Radon–Nikodym derivatives are well-defined. Second, $x \mapsto \eta_\theta(x)$, $x \mapsto A(\eta_\theta(x))$, $x \mapsto \mu_\theta(x)$, and $x \mapsto s_\theta(x; y)$ must be measurable. Third, the normalizer must be finite and nonzero for the observations under consideration. Fourth, $A$ should be closed, strictly convex, and sufficiently smooth on the relevant natural-parameter region, with smooth Legendre dual $A^*$ on the corresponding mean-parameter region. Fifth, tail and integrability conditions must justify limit interchanges and control posterior mass outside compact sets. Finally, the finite softmax stability and log-odds converse would need density-level analogues: bounded log-density perturbations can control total variation or Hellinger distance, while finite log-odds vectors become centered log Radon–Nikodym derivatives in an appropriate $L^2$ space. These function-space steps are outside the scope of the present finite-class paper.

