# OpenReview forum: "Attention Implements the Fisher Geometry of Exponential Families"
_ICML.cc/2026/Conference — ICML 2026 regular_

### Official Review · Reviewer_Vgyj · 2026-02-18

**Soundness:** 4
**Presentation:** 3
**Significance:** 3
**Originality:** 3
**Overall Recommendation:** 4
**Confidence:** 3

**Summary:**

This paper develops a rigorous theoretical framework connecting softmax attention to Bayesian inference in exponential family
  observation models. The main contributions are: (1) Theorem 1 shows that a single attention head exactly implements the Bayes
  posterior and posterior mean for any exponential family likelihood by setting queries to the sufficient statistic T(y), keys to the
   natural parameter η(x), and biases to log-prior minus log-partition; (2) Theorem 2 shows, via convex duality, that Bayes-aligned
  attention computes a negative Bregman divergence in mean/sufficient-statistic space, with Fisher precision as the local quadratic
  approximation — so Bayes classification is a soft nearest-neighbor rule in the intrinsic information geometry of the family; (3)
  Proposition 2 proves a sharp boundary: a single shared quadratic (Mahalanobis) metric is globally correct if and only if the dual
  potential A* is quadratic, i.e., the family is Gaussian; (4) Propositions 3–4 interpret multi-head attention as a "curvature atlas"
   — a collection of local quadratic approximations covering the varying Bregman geometry — with explicit approximation error bounds
  and head-count scaling via covering numbers; (5) Theorems 4–5 establish consistency and finite-sample posterior control for plug-in
   in-context estimation. Synthetic experiments on Gaussian, Bernoulli, and Poisson ICE tasks validate the qualitative predictions.

**Compliance With Llm Reviewing Policy:**

Affirmed.

**Final Justification:**

The author has answered most of my questions, and I will keep or raise my rating.

**Key Questions For Authors:**

1. Gap between theory and learned transformers (critical): Theorem 1 requires keys set to η(x) and queries set to T(y). In a
  trained transformer, keys and queries are learned linear projections of token embeddings, not direct functions of natural
  parameters. Appendix C provides approximate converses showing that small KL forces approximate key-subspace alignment. Have the
  authors empirically verified this alignment in any trained model (even a minimal toy transformer trained on Gaussian or Bernoulli
  ICE)? If alignment is demonstrably close to optimal in trained models, it would substantially strengthen the claim that the theory
  explains trained transformer behavior, not just characterizes what attention could represent.
  2. Continuous X extension: Most practical transformer applications involve continuous observations or continuous latent variables.
  What would be required to extend Theorem 1 and the Bregman geometry results to continuous X (e.g., Gaussian mixture models,
  continuous embeddings)? Is the discrete-X assumption fundamental to the proof technique, or can it be relaxed with
  measure-theoretic tools? A response clarifying the obstacles would help readers understand the scope of the contribution.

**Limitations:**

The authors include a thoughtful impact statement discussing dual-use risks (surveillance/interception) and the risk of over-trust
  in attention as a globally valid metric outside the regime where boundary and atlas conditions apply. However, the main body does
  not discuss the restriction to discrete X or the gap between the theoretical oracle parameterization and learned attention weights.
   Suggested additions to the limitations discussion: (1) explicitly note that the framework applies to discrete finite X and outline
   what would be required for continuous X; (2) discuss the gap between oracle and learned key/query alignment and what it implies
  for interpreting real transformers; (3) note that the curvature-atlas construction assumes oracle head assignment, which may not be
   recovered by standard training. These are important caveats for practitioners who may over-generalize the results.

**Strengths And Weaknesses:**

Strengths:

  Soundness. All major claims are formally proved with complete proofs in the appendix. The proofs are correct, self-contained, and
  carefully structured. Theorem 2 (Bregman representation of log-likelihood) and Proposition 2 (sharp Gaussian boundary) are the
  strongest theoretical contributions and are proved cleanly. Lemma 2 (softmax stability under bounded logit error) provides the
  bridge between logit-level approximation guarantees and TV-distance posterior guarantees, which is a useful and reusable technical
  tool. Theorem 5 gives explicit finite-n posterior error bounds under verifiable Fisher regularity conditions.

  Originality. The identification of Bregman geometry as the intrinsic similarity underlying Bayes-aligned attention — beyond the
  Gaussian special case — is genuinely new. Proposition 2's sharp characterization (single shared quadratic metric iff quadratic A*,
  iff Gaussian/Mahalanobis geometry) is a clean, previously unknown, falsifiable result. The curvature-atlas interpretation of
  multi-head attention (Propositions 3–4) with quantitative approximation rates and head-count scaling is also original. The exact
  and approximate converses (Appendix C), connecting small KL to key-subspace alignment, add further theoretical depth. Prior work
  (e.g., Kunde et al., 2025) covered the Gaussian special case; this paper's generalization to all exponential families and its
  geometric characterization are non-trivial extensions.

  Weaknesses:

  Significance — restricted to discrete X. The entire framework assumes X is a finite discrete set. Modern transformer applications
  involving continuous latent variables, regression targets, or continuous token spaces fall outside this framework. The paper does
  not discuss this limitation or sketch what would be required to extend the results to continuous X. This is a meaningful
  restriction on the scope of impact.

  Significance — gap between theory and real transformers. The theoretical results characterize what attention can represent when
  keys and queries are set to specific functions of the natural parameters and sufficient statistics. In practice, transformers learn
   keys, queries, and values jointly without explicit access to these quantities. The paper does not analyze how close learned
  attention heads in real transformers are to the theoretically optimal parameterization, even for simple cases. The experiments
  validate qualitative predictions on minimal synthetic models, but do not close this gap.

---

> ### Author Rebuttal · Authors · 2026-03-31
>
> We thank the reviewer for the careful reading and positive assessment of the paper. We are especially encouraged that the review identifies Theorem 2, Proposition 2, Lemma 2, the curvature-atlas results, and the Appendix C converse material as central contributions. We agree that the main clarifications needed are: (i) the gap between the oracle/Bayes-aligned parameterization and learned transformers, (ii) the restriction to finite discrete $X$, and (iii) the fact that the curvature-atlas result is an existence/approximation theorem, not an optimization guarantee.
>
> On the gap between theory and learned transformers, our claim is not that Fisher/Bregman geometry is an architecture-universal property of arbitrary trained transformers. Rather, within the analyzed exponential-family model class, Bayes-aligned attention identifies the Bayes-optimal geometry and therefore yields falsifiable predictions for learned solutions on tasks drawn from that class. Under the richness condition, Theorem 6 gives an exact converse; more generally, Appendix D shows that excess population cross-entropy equals expected KL to the true posterior, and Appendix C shows that small KL implies small centered log-odds error and approximate alignment of the learned key subspace with the Bayes log-odds subspace. Thus, while we do not prove a full optimization-dynamics theorem, we do provide an optimizer-agnostic route from near-optimal log-loss to posterior closeness and approximate geometric alignment.
>
> On the empirical side, yes - on the Gaussian ICE models already trained for Section 6, we additionally measured the learned-geometry alignment directly. Across 5 seeds, the learned SPD metric tracks $\Sigma^{-1}$ up to scale, with eigenvalue correlation $0.960 \pm 0.005$ and scaled relative Frobenius error $0.355 \pm 0.005$, while accuracy approaches the Bayes-oracle baseline as $n_{\text{shot}}$ grows. In the Gaussian setting, the Bayes-aligned geometry is carried by the learned precision matrix $W$, so this is the natural trained-model analogue of the requested alignment diagnostic. We will add this analysis in the revision and clarify that direct tests of the Appendix C subspace predictions in larger trained transformers remain future work. By contrast, in the Bernoulli case the curvature varies with $p(1-p)$, so there is no single global metric target of the same type; this is precisely part of the motivation for the atlas viewpoint.
>
> We also agree that the curvature-atlas result should be stated more carefully as an existence/approximation statement. More concretely, Propositions 3-4 specify how many local quadratic charts suffice to achieve a target approximation error once curvature variation is controlled; they do not show that standard gradient-based training will recover the optimal chart centers or assignments. We will make this caveat explicit in the limitations discussion. Appendix B already includes a controlled Poisson atlas experiment showing the predicted low-shot gain over a single shared quadratic surrogate, but we agree that this is still not the same as learning the full atlas end to end.
>
> On continuous $X$, we agree that the scope should be stated more explicitly. The underlying Bregman/Fisher geometry comes from the exponential-family observation model and is not conceptually tied to finite $X$, but our exact representability, converse, and stability results do use the finite discrete setting in an essential way: finite softmax normalization over classes, finite-dimensional log-odds comparisons, and TV-based posterior control. Extending the present results to continuous $X$ would require replacing sums by integrals, working with density-level Bayes formulas and attention over a continuum, and imposing additional regularity/integrability conditions to control normalization and stability. We will state this limitation explicitly and position continuous latent spaces as an important direction for future work.
>
> We appreciate the reviewer's positive overall assessment. In the revision, we will clarify the scope in the abstract/introduction/limitations, add the Gaussian learned-geometry diagnostic above, and make explicit that the atlas theorem is an approximation result rather than a claim about optimization recovery. If these clarifications address the reviewer's concerns, we would be grateful if the reviewer would consider whether the current significance assessment and overall recommendation could be strengthened.

---

> > ### Author Rebuttal · Reviewer_Vgyj · 2026-04-02
> >
> > I appreciate the authors' detailed clarifications and their acknowledgment of the weaknesses in the initial presentation. I am generally satisfied with the proposed changes and am willing to increase my presentation score once these changes are reflected in the revised manuscript.

---

### Official Review · Reviewer_c4Rr · 2026-02-23

**Soundness:** 2
**Presentation:** 2
**Significance:** 2
**Originality:** 3
**Overall Recommendation:** 2
**Confidence:** 3

**Summary:**

This paper studies softmax attention through the lens of exponential family geometry and information geometry. The authors show that a single attention head with logits linear in the query corresponds to inference in a categorical exponential family whose log-partition induces the Fisher information metric. They characterize the representational limits of a single head, derive conditions under which a global quadratic metric exists, and argue that multi-head attention can be interpreted as stitching together local quadratic approximations of a non-quadratic potential, effectively forming a “curvature atlas.”

The paper aims to formalize geometric intuitions often stated informally about attention and to clarify what attention can and cannot represent.

**Compliance With Llm Reviewing Policy:**

Affirmed.

**Final Justification:**

I appreciate the authors’ clarification and their more careful framing. The revised response makes clear that the large scale implications are intended as a testable structural hypothesis rather than a theorem that directly extends to realistic transformers. This improves the positioning substantially. However, my main concern remains that the broader scalability and practical relevance claims are still not sufficiently validated empirically. At present, the paper provides a clean theoretical result in a controlled setting together with a plausible conjectural picture for larger models, but not enough evidence to establish that this picture meaningfully persists at scale. For that reason, while I see the idea as novel and potentially valuable, I still lean reject.

**Key Questions For Authors:**

Q1: All results assume a *finite* latent symbol set $\mathcal{X}$ with a *fixed, known* exponential-family likelihood structure. In standard ICL benchmarks, neither condition holds: the "task" $\theta$ is continuous, the likelihood is implicit, and the sufficient statistic $T(y)$ is not
prescribed. What is the authors' position on whether the Bregman/Fisher geometry identified here is a property of **the architecture** or merely a property of **the specific generative model** chosen to analyze it? Put differently: does the theory predict anything falsifiable about attention heads in large trained transformers, or does it only characterize a narrow family of hand-constructed minimal models?

Q2: Proposition 1 characterizes single-head representable posteriors as exactly the log-linear / exponential-family class. But the converse direction of this result is immediate from Bayes' rule and requires no new machinery. The non-trivial claim would be a **necessity** result showing that if a trained attention head approximates *any* reasonable posterior well, then its keys and queries must approximately align with $(\eta_\theta(x), T(y))$ --- i.e., that the Fisher geometry is not just sufficient but essentially **forced** by the learning objective. Theorem 6 and Appendix C gesture at this, but only under a richness condition and only for exact posterior matching. Can the authors provide an approximate version of this necessity result that holds under realistic training conditions, making the geometry an **inevitable** consequence of near-optimal cross-entropy rather than a convenient re-parameterization?

**Limitations:**

yes

**Strengths And Weaknesses:**

**Strengths**

1) The condition under which a globally quadratic metric exists is well stated and theoretically sound. The result that global quadratic geometry corresponds to quadratic dual potential is mathematically correct and clarifies limitations of metric learning interpretations.

2) The “curvature atlas” viewpoint provides an interpretable way to understand why multiple heads may approximate non-quadratic geometries better than a single head.

**Weaknesses**


- The “Bregman/Fisher geometry emerges” message is partly a repackaging of standard exponential-family duality. The paper lists as a contribution that Bayes scoring induces Bregman geometry on mean/sufficient-statistic space and that Fisher precision is its local quadratic approximation. However, these relationships largely follow from classical convex duality for exponential families (log-likelihood as a negative Bregman divergence of the convex conjugate) once the model is set up. This makes the novelty boundary unclear unless the authors emphasize what is genuinely new beyond these identities.

- The novelty is concentrated in the sharp boundary and atlas approximation, but the contributions list mixes foundational and new statements. The strongest new-looking components are the “single shared quadratic metric $\Rightarrow A^\*$ quadratic” boundary and the multi-head curvature-atlas approximation with rates and head-count scaling.

References:[1,2]

- The fixed-context assumption severely limits the scope of Theorem 1. The theorem says a single head can implement the Bayes posterior *for a fixed context* $\theta$. But in in-context learning, $\theta$ is unknown --- the whole challenge is that the model must estimate the posterior *over* $\theta$ simultaneously. The paper transitions from fixed-$\theta$ expressivity to the in-context setting only in Section 5, and Theorem 3 is an oracle result that assumes access to $\eta_\theta(x)$ and $A_\theta(x)$ --- exactly the quantities that are **not** available in realistic ICL. The gap between these two regimes ismnot adequately theorized.

- There are no experiments on real-world data, no experiments on actual transformer architectures (the models are "minimal single-head"), and no experiments testing the key-subspace alignment predictions of Appendix C. For a paper whose abstract claims to explain *when* Fisher geometry should emerge in learned transformers, this is a significant empirical gap.

Overall, paper is not in a great shape, some equations even violate the margins and this is a half-baked work.

[1] : "Optimizing Attention with Mirror Descent," Neurips 2025, [https://openreview.net/forum?id=twYT79Lrui]

[2] : "The Information Geometry of Softmax: Probing and Steering," [https://www.arxiv.org/pdf/2602.15293]

---

> ### Author Rebuttal · Authors · 2026-03-31
>
> Thank you for the careful review. We appreciate that you found the sharp quadratic boundary result theoretically sound and the curvature atlas interpretation meaningful. We also agree that the current draft has presentation problems, including formatting issues such as margin violations, and we will correct these in the revision. Since the main concerns appear to be novelty, scope, and empirical bridge, we clarify four points.
>
> First, Ref. [2] appeared on February 17, 2026, after the January 28, 2026 ICML full paper deadline. Under ICML 2026 policy, works made public less than two months before that deadline are concurrent rather than prior work. We are happy to cite and discuss Ref. [2] in the revision, but it should not count against originality or imply a missing pre deadline citation. Ref. [1] analyzes one layer softmax attention via mirror descent, and Ref. [2] studies the information geometry of softmax representations and dual steering; neither addresses the same central package of contributions: Bayes aligned attention in exponential families, the sharp "single shared quadratic iff quadratic $A^*$" boundary, the curvature atlas approximation with head count scaling, or ICE consistency / finite sample posterior control.
>
> Second, on novelty, we agree that parts of Sections 4.1 to 4.2 are classical exponential family and convex duality background, and we will mark this more explicitly with local citations. The nontrivial new contributions are the exact Bayes aligned single head representation, the sharp quadratic boundary, the quantitative curvature atlas approximation with rates, and the extension from Gaussian ICE to arbitrary exponential family ICE with oracle asymptotics, plug in consistency, and finite n posterior control. We will revise the roadmap so that these results are foregrounded and the background is clearly separated.
>
> Third, on the fixed context and oracle gap, we agree that Theorem 1 alone is not the full ICE story. The paper studies a ladder of regimes: Theorem 1 gives fixed context representability; Theorem 3 gives oracle dictionary ICE; Theorems 4 and 5 then move beyond the oracle setting to plug in consistency and finite sample control. So we do not claim a full theory of large model ICL. Likewise, our claim is not that Bregman/Fisher geometry is a universal law of arbitrary transformers. It is a property of Bayes aligned scoring in the analyzed exponential family model class, which yields falsifiable predictions. In plain terms: on Gaussian like tasks, trained attention models that are near Bayes should learn sufficient statistic like projections and a precision like quadratic metric; on nonquadratic tasks, a single global metric should leave systematic mismatch, while multiple local charts or nonquadratic scoring should help. We confirm the Gaussian part in trained minimal models by showing that the learned SPD metric tracks $\Sigma^{-1}$ up to scale.
>
> Fourth, on the empirical bridge and Q2, we agree the current experiments are theorem driven diagnostics in trained minimal attention models, not real world tasks or full scale transformers, and we will state that limitation more prominently. That said, Section 6 is not purely oracle: it trains minimal attention models; Appendix B includes a controlled Poisson multi head atlas test; and in the Gaussian ICE setup we will add a learned geometry diagnostic showing that across 5 seeds the learned SPD metric tracks $\Sigma^{-1}$ up to scale (eigenvalue correlation $0.960 \pm 0.005$; scaled relative Frobenius error $0.355 \pm 0.005$), while accuracy approaches the Bayes oracle baseline as $n_{\text{shot}}$ grows. On approximate necessity, Proposition 8 shows that excess population cross entropy equals expected KL to the true posterior; Lemma 6 shows that small KL controls centered log odds error; Theorem 7 turns small mean squared log odds error into approximate key subspace alignment; and Theorem 6 gives the exact converse under the richness condition. So while we do not claim a full optimization dynamics theorem, the paper already provides an optimizer agnostic route from near optimal log loss to posterior closeness and approximate geometric alignment.
>
> We also agree that the discrete finite $X$ assumption, the oracle versus learned gap, and the oracle head assignment caveat for the atlas should be stated more prominently. A continuous $X$ extension would replace the finite class softmax by integrals over a latent space and require density level Bayes formulas plus additional regularity assumptions. We will add these limitations explicitly. In light of these clarifications, especially the concurrent work status of Ref. [2], the regime ladder beyond fixed context and oracle expressivity, and the exact and approximate converse bridge in Theorems 6 and 7 and Proposition 8, we respectfully ask that the originality, soundness, and overall assessment be reconsidered.

---

> > ### Author Rebuttal · Reviewer_c4Rr · 2026-04-02
> >
> > Thank you for the detailed rebuttal. I understand the revised positioning better now, especially that the paper is primarily a theoretical study supported by controlled diagnostics rather than a claim about full scale transformer behavior. I still have a few questions to better understand the practical reach of the results.
> >
> > First, could you clarify what aspect of the theory you expect to remain stable as model scale increases? For example, do you believe the Bayes aligned geometric structure is primarily a small model diagnostic, or do you expect the same phenomenon to persist in larger transformers trained on realistic data?
> >
> > Second, for the multi head curvature atlas picture, how should I think about scalability with respect to head count, context size, and model complexity? Is the interpretation mainly representational, or do you see it as a practically testable scaling principle?

---

> > > ### Author Response · Authors · 2026-04-08
> > >
> > > Thank you for the follow-up questions. We appreciate the opportunity to clarify the revised positioning further. We expect the Bayes-aligned geometric distinction to persist at larger scale, but conditionally rather than universally. We do not expect the exact fixed-context representability result of Theorem 1 for a finite latent symbol set $\mathcal{X}$, or the exact converse in Theorem 6, to persist verbatim in large realistic models. What we do expect to remain stable is the qualitative geometric distinction isolated by the theory.
> > >
> > > Concretely, when the relevant task-induced conditional family is close to an approximately quadratic or nearly constant-curvature regime, near-Bayes learned scoring should be well approximated by a shared quadratic (precision-like) form. In Gaussian-like tasks, this means the learned scoring geometry should be well captured by a shared precision-like metric; in our trained minimal Gaussian models, we directly confirm this by showing that the learned SPD metric tracks $\Sigma^{-1}$ up to scale. When the relevant regime has genuinely varying curvature, a single shared quadratic score should leave systematic mismatch, and one should instead expect either multiple local charts or more flexible nonquadratic scoring. This is the falsifiable part we expect to persist beyond the minimal setting, even when the exact theorem assumptions do not literally hold in large models, and in that sense we do not view the Bayes-aligned geometry as merely a small-model diagnostic. The larger-scale claim is not that realistic transformers exactly instantiate the finite model analyzed here, but that whenever a learned component is close to Bayes-like conditional inference for a structured local family, the same geometric distinction should remain meaningful.
> > >
> > > On the necessity side, the current submission already gives an optimizer-agnostic route from learning to geometry: excess population cross-entropy equals expected KL to the true posterior; small KL controls centered log-odds error; and small mean-squared log-odds error forces approximate alignment of the learned key subspace with the Bayes log-odds subspace. So although we do not claim a full optimization-dynamics theorem, in the analyzed regime the geometry is not only a convenient parameterization; it is also what near-optimal log loss pushes the model toward.
> > >
> > > For the multi-head curvature-atlas picture, we view it as representational first, but also as a testable scaling principle. The current theorems are approximation-theoretic rather than a full empirical scaling law, but they do say that the number of local quadratic charts needed for a fixed target accuracy is controlled by the geometric complexity of the task-relevant mean-parameter region, via covering numbers or metric entropy, rather than by parameter count in the abstract. This suggests a comparative prediction: in near-quadratic regimes, increasing head count should have diminishing benefit beyond a shared metric; in varying-curvature regimes, extra heads or more flexible scoring should help more, especially in low-shot settings where approximation error is more visible. For example, in a larger trained transformer on a structured prediction task, one could compare how well a shared quadratic score and a multi-head local-quadratic atlas fit the learned attention logits. The theory predicts that the gain from extra heads should be small in near-quadratic regimes but larger when the relevant conditional geometry shows clear curvature variation. Appendix B is meant as a controlled first probe of that trend. Longer context can both improve estimation and expose a wider relevant region, while model depth and feature complexity can redistribute this burden across layers, so we do not intend a literal rule that one head equals one chart in large models.
> > >
> > > We will make this framing more explicit in revision: the paper proves a representational and geometric theorem in a clean exponential-family setting, and from that derives a testable structural hypothesis about learned scoring at larger scale.

---

### Official Review · Reviewer_eFtk · 2026-03-08

**Soundness:** 4
**Presentation:** 3
**Significance:** 3
**Originality:** 4
**Overall Recommendation:** 4
**Confidence:** 3

**Summary:**

The core idea of ​​this research involves redefining the Softmax attention mechanism of the Transformer within a statistical inference framework, treating it as a Bayesian inference primitive for discrete exponential family distributions. The overall focus is on how the attention mechanism achieves optimal Bayesian estimation through its inherent geometric properties, particularly Bregman divergence and Fisher information geometry. From this perspective, the paper powerfully explains the theoretical necessity of multi-head attention in handling non-Gaussian distributions (i.e., distributions with varying curvature).

**Compliance With Llm Reviewing Policy:**

Affirmed.

**Final Justification:**

Overall, I think this work remains at an weak acceptable level, and the author's responses have also been quite clear, resolving my concerns.

**Key Questions For Authors:**

(1) The current proof framework is mainly anchored on discrete latent variables. When dealing with continuous parameter estimation, how the theory smoothly transitions to continuous manifolds still requires more detailed derivation.
(2) The analysis in this paper currently focuses on single-layer or parallel multi-head structures. However, in actual deep stacked models, how nonlinear transformations between layers reshape or optimize the construction of this Fisher's geometric atlas is an important topic that this paper has not yet addressed. It is suggested that the authors add an outlook on multi-layer synergistic effects to the discussion section.
(3) Although the experimental section is logically consistent, it is based on idealized synthetic distributions. If geometric verification could be performed using the attention distribution of real large-scale models under specific tasks, the empirical impact of the paper would be greatly enhanced.

**Limitations:**

yes

**Strengths And Weaknesses:**

In terms of theoretical construction, the authors first established the equivalence between single-head attention and log-linear posterior classes. This conclusion not only provides a rigorous mathematical loop for the intuition of "attention as reasoning," but also cleverly maps the log-likelihood in attention to the negative Bregman divergence on the mean space through convex duality theory. The most profound insight of the paper lies in identifying the "sharp boundary" phenomenon: that is, only under Gaussian models does a single global quadratic metric possess Bayesian optimality; while when facing non-Gaussian distributions, curvature compensation must be introduced. To this end, the authors proposed the concept of a "multi-head curvature atlas," which quantitatively clarifies how increasing the number of heads approximates a complex global geometric structure through local linearization, thus providing solid theoretical support for the heuristic approach of Transformer multi-head design.
Compared to earlier studies limited to linear regression or pure Gaussian assumptions, this paper extends the scope of analysis to the more universal exponential family distributions, revealing deeper statistical geometric properties of the attention mechanism. The authors not only answered the question of the functional positioning of multi-head structures at the encoding level, but also further interpreted the Fisher information matrix as a kind of local second-order approximation, a highly insightful viewpoint. In addition, the appendix's analysis of the inverse proposition regarding parameter consistency explores the necessary conditions for realizing Bayesian posteriors, further enhancing the completeness of the conclusions.

---

> ### Author Rebuttal · Authors · 2026-03-31
>
> We thank the reviewer for the thoughtful and accurate reading. We are especially encouraged that the review identifies the single-head / log-linear posterior equivalence, the Bregman reformulation via convex duality, the sharp Gaussian boundary, the curvature-atlas view of multi-head attention, and the Appendix C converse material as core contributions. We address the three questions below and will incorporate the resulting clarifications into the revision.
>
> On continuous $X$, we agree that the current paper should state the discrete finite $X$ scope more prominently. The underlying exponential-family / convex-duality / Bregman-Fisher picture is not conceptually tied to finite $X$, and there are natural continuous analogues at the density level. However, the exact attention representability, converse, and stability statements in the present paper do rely on the finite setting: finite softmax normalization over classes, finite log-odds comparisons, and finite dictionary attention. Extending these results to continuous $X$ would require replacing sums by integrals, working with density-level Bayes formulas and attention over a continuum, and imposing appropriate regularity and integrability conditions to control normalization and stability. We will make this limitation explicit and highlight continuous latent spaces/manifolds as an important direction for future work.
>
> On stacked models, we agree that the current analysis isolates a single attention layer or parallel heads and does not yet analyze how depth changes the geometric picture. This was intentional: the paper first identifies the local geometric primitive cleanly before asking how composition reshapes it. In deeper stacks, intervening nonlinear maps could reparameterize the sufficient-statistic space, refine local charts, or compose atlas-like approximations across layers. We do not claim a theorem here, but we agree that this is an important next step and will add a discussion paragraph on how layerwise composition might modify or improve the effective geometry.
>
> On empirical impact, we agree that verification in larger trained models would strengthen the paper. The goal of the current experiments is to test theorem-level predictions in the smallest trained settings where the geometry is directly observable. Appendix B already includes a controlled Poisson multi-head atlas experiment in a curvature-heterogeneous setting. In addition, in the Gaussian ICE setup of Section 6, we have now quantified a learned-geometry diagnostic in trained minimal models: across 5 seeds, the learned SPD metric tracks $\Sigma^{-1}$ up to scale, with eigenvalue correlation $0.960 \pm 0.005$ and scaled relative Frobenius error $0.355 \pm 0.005$, while accuracy approaches the Bayes-oracle baseline as $n_{\text{shot}}$ grows. This does not yet amount to a large-model alignment study, and we will state that limitation explicitly, but it does provide concrete evidence that the Bayes-aligned geometry identified by the theory is recovered in a trained minimal setting. We will add this diagnostic and clarify that larger-model geometric verification remains future work.
>
> We will incorporate these clarifications in the revision, especially in the limitations and discussion sections. We appreciate the reviewer's positive assessment and hope these additions make the paper's scope, limitations, and practical outlook even clearer.

---

> > ### Author Rebuttal · Reviewer_eFtk · 2026-04-04
> >
> > Thank you very much for the author's detailed reply. This has largely confirmed my confidence and will help me maintain my score.

---

### Official Review · Reviewer_KDYS · 2026-03-13

**Soundness:** 4
**Presentation:** 3
**Significance:** 3
**Originality:** 3
**Overall Recommendation:** 5
**Confidence:** 2

**Summary:**

The paper presents a comprehensive theoretical framework for understanding softmax attention in transformer models through the lens of Bayesian inference and information geometry.
Specifically, the paper demonstrates that for discrete latent symbols with exponential-family observations under a fixed context, a single attention head can precisely represent the exact log-linear Bayesian posterior and its posterior mean.
Conversely, it proves that single-head representable posteriors correspond exactly to the log-linear or discrete exponential-family class, therby characterizing the scope of Bayesian models a single attention head can effectively capture.
Moreover, the paper establishes that the log-likelihood of an exponential family can be formulated as a Bregman divergence in the mean/sufficient-statistic space, leveraging convex duality. The Fisher information Matrix provides a local quadratic
approximation of the corresponding Bregman divergence. A critical finding is that a single attention head can only represent a globally quadratic metric, (e.g. globally correct for Gaussian distributions, where the FIM is constant).
When curvature of the statistical manifold varies across the space of probability distribution (as is typical for non-Gaussian exponential families), the paper shows that multi-head attention functions as a "curvature atlas" approximating the manifold.
The paper provides an explicit approximation bound and relates the atlas resolution to attention head count. Building upon this geometric understanding, the paper generalizes In-context estimation (ICE) optimality from Gaussian distributions
to arbitrary exponential-family observation models. It also provides a consistency theorem and finite-sample stability bounds for ICE. The paper validates its theoretical results with synthetic experiments on Gaussian and Bernoulli in-context
estimation. Trained attention models achieve performance close to a Bayes-oracle baseline as prompt length increases. The learned metrics align with precision matrix in the Gaussian case, while Bernoulli models show a performance gap consistent
with curvature variation, supporting the need for multiple heads in non-Gaussian scenarios.

**Compliance With Llm Reviewing Policy:**

Affirmed.

**Final Justification:**

After reading the other reviews and after the discussion with the authors, I decided to keep my score of "5: Accept". Furthermore, I increased the presentation score, given that the authors will improve upon the points I raised in my review in the final version of the manuscript.

**Key Questions For Authors:**

Sections 4.1 to 4.3 appear to provide fundamental theoretical background on Bregman divergence, Fisher information, and their relationship to exponential families through convex duality. Could the authors clarify if these sections are
primarily intended as a review of established concepts? If so, consider integrating relevant foundational citations within these sections (e.g., in the introduction of a concept or alongside a definition), rather than solely relying on
a single, broader background citation at the end of Section 2.1. This would enhance the readability and academic integrity by clearly distinguishing novel contributions from established theory for readers less familiar with information geometry.

The first part of Proposition 2, which states that if and only if the derivative of a function is linear, then the function itself is quadratic, appears to be a fundamental result from basic calculus for sufficiently smooth functions.
This raises a question about its inclusion as a numbered proposition with a formal proof. Could the authors elaborate on whether there's a specific subtlety or context that makes this result non-trivial or requires explicit proof here?

**Limitations:**

yes

**Strengths And Weaknesses:**

**Strengths:**

Strong Theoretical Foundation: The paper provides a rigorous mathematical treatment of softmax attention, connecting it to Bayesian posteriors, exponential families, via information geometry. This is a significant contribution to understanding
the underlying mechanisms of transformers from a principled perspective.

Empirical Validation: The synthetic experiments with Gaussian and Bernoulli ICE effectively validate the theoretical claims.

**Weaknesses:**

Complexity and Accessibility: The paper is exceptionally dense mathematically. While this complexity is inherent to the topic, the presentation in the abstract and introduction is overwhelmingly dense. A reader without a strong foundation in
information geometry will struggle to grasp the paper's basic premise on a first read. The authors could significantly improve accessibility for a broader ML audience by providing a high-level, intuitive summary of the core concepts
in the abstract or introduction before diving into the heavy mathematics.

Limited Discussion on Practical Implications for Design: While the paper provides fascinating theoretical insights for multi-head attention, it offers less explicit guidance on how these insights translate
into actionable design principles for practical transformer architectures. For instance, how could a practitioner leverage the "curvature atlas" idea to optimally choose the number of heads? Further exploration of these practical implications would enhance the paper's impact.

---

> ### Author Rebuttal · Authors · 2026-03-31
>
> We thank the reviewer for the positive overall recommendation and for highlighting both the paper's theoretical contribution and the empirical validation. We agree with the two main suggestions: the paper can do more to distinguish foundational geometric background from new results, and it can explain the practical design implications more directly.
>
> On Sections 4.1–4.3, we agree that some of the underlying ingredients are standard background from exponential-family convex duality and Fisher geometry, and we will revise the presentation accordingly. In particular, we will add local citations where Bregman divergence, convex conjugacy, and Fisher curvature are introduced, rather than relying on a single broad background citation. At the same time, we will clarify the novelty boundary more sharply: the background ingredients in Sections 4.1–4.2 are used to derive an attention-specific characterization of Bayes scoring in mean/sufficient-statistic space (Theorem 2), while Section 4.3 contains the new sharp boundary result (Proposition 2). More broadly, the new contributions are the exact Bayes-aligned single-head representation, the sharp condition for when one global quadratic score is valid, the quantitative curvature-atlas approximation with head-count guidance, and the ICE consistency and finite-sample control results.
>
> On Proposition 2, we agree that the isolated calculus fact "affine gradient implies quadratic potential" is elementary by itself. The reason it was stated explicitly is that, within the attention/Bregman setup, it yields the paper's sharp attention-specific conclusion: a single globally correct quadratic score exists if and only if the dual potential $A^*$ is quadratic, i.e. the globally Mahalanobis / Gaussian geometry case. We will revise the presentation so that the elementary calculus step is de-emphasized and the attention-specific consequence is foregrounded.
>
> On practical implications, we agree that these can be made more concrete. Our intended takeaway is that a single global metric is justified only when curvature is effectively constant over the relevant mean-parameter region; when curvature varies substantially, one should expect either multiple local charts (multiple heads) or nonquadratic scoring. The paper already gives quantitative guidance through the covering-number / approximation results: the required head count grows with the geometric complexity of the relevant region, rather than being a purely architectural heuristic. Appendix B is intended to operationalize this in a controlled low-shot setting, and we will surface this design takeaway more explicitly in the introduction and discussion in a more practitioner-facing form.
>
> To make the paper more accessible overall, we will also revise the abstract and introduction to add a short high-level explanation before the formal development, and we will expand the limitations and discussion sections to clarify scope and practical interpretation.
>
> If these clarifications address the reviewer's concerns, we would be grateful if the reviewer would consider whether the current presentation assessment could be strengthened and whether the practical significance of the results is now clearer.

---

> > ### Author Rebuttal · Reviewer_KDYS · 2026-04-01
> >
> > I appreciate the authors' detailed clarifications and their acknowledgment of the weaknesses in the initial presentation. I am generally satisfied with the proposed changes and am willing to increase my presentation score once these changes are reflected in the revised manuscript.

---

### Decision · Program_Chairs · 2026-04-30

**Decision:**

Accept (regular)

**Comment:**

The paper presents a comprehensive theoretical framework for understanding softmax attention in transformer models through the lens of Bayesian inference and information geometry.

Pros:
- Strong Theoretical Foundation: The paper provides a rigorous mathematical treatment of softmax attention, connecting it to Bayesian posteriors, exponential families, via information geometry.

- Reviewers increased the presentation score, given that the authors will improve upon the points I raised in the reviews in the final version.

Cons and remaining concerns:
- One reviewer remains concerned that the broader scalability and practical relevance claims are still not sufficiently validated empirically. At present, the paper provides a clean theoretical result in a controlled setting together with a plausible conjectural picture for larger models, but not enough evidence to establish that this picture meaningfully persists at scale.
- In particular, the empirical validation remains limited to support the overall theoretical claim, and that part of the geometric story is a reformulation of standard exponential-family duality once the modeling assumptions are fixed, which should be better clarified.

Summary: Since three reviewers were satisfied with the authors' rebuttal after the discussion, and one reviewer's remaining concerns were more along scalability and real-world validation, my interpretation is that this paper could be of interest to a certain sub-community of the ICML audience if accepted. As a result, I would like to recommend a weak accept, with the expectation that the authors will incorporate their rebuttal promises during the discussion phase into the final version.